# Isolation and Characterization of Plant-Growth-Promoting, Drought-Tolerant Rhizobacteria for Improved Maize Productivity

**DOI:** 10.3390/plants13101298

**Published:** 2024-05-08

**Authors:** Victor Funso Agunbiade, Ayomide Emmanuel Fadiji, Nadège Adoukè Agbodjato, Olubukola Oluranti Babalola

**Affiliations:** Food Security and Safety Focus Area, Faculty of Natural and Agricultural Sciences, North-West University, Mmabatho 2735, South Africa

**Keywords:** plant-microbial interaction, sustainable agriculture, 16S rRNA gene, drought tolerance, biological control, stress condition

## Abstract

Drought is one of the main abiotic factors affecting global agricultural productivity. However, the application of bioinocula containing plant-growth-promoting rhizobacteria (PGPR) has been seen as a potential environmentally friendly technology for increasing plants’ resistance to water stress. In this study, rhizobacteria strains were isolated from maize (*Zea mays* L.) and subjected to drought tolerance tests at varying concentrations using polyethylene glycol (PEG)-8000 and screened for plant-growth-promoting activities. From this study, 11 bacterial isolates were characterized and identified molecularly, which include *Bacillus licheniformis* A5-1, *Aeromonas caviae* A1-2, *A. veronii* C7_8, *B. cereus* B8-3, *P. endophytica* A10-11, *B. halotolerans* A9-10, *B. licheniformis* B9-5, *B. simplex* B15-6, *Priestia flexa* B12-4, *Priestia flexa* C6-7, and *Priestia aryabhattai* C1-9. All isolates were positive for indole-3-acetic acid (IAA), siderophore, 1-aminocyclopropane-1-carboxylate (ACC) deaminase, ammonia production, nitrogen fixation, and phosphate solubilization, but negative for hydrogen cyanide production. *Aeromonas* strains A1-2 and C7_8, showing the highest drought tolerance of 0.71 and 0.77, respectively, were selected for bioinoculation, singularly and combined. An increase in the above- and below-ground biomass of the maize plants at 100, 50, and 25% water-holding capacity (WHC) was recorded. Bacterial inoculants, which showed an increase in the aerial biomass of plants subjected to moderate water deficiency by up to 89%, suggested that they can be suitable candidates to enhance drought tolerance and nutrient acquisition and mitigate the impacts of water stress on plants.

## 1. Introduction

The impact of abiotic-induced drought stress on plant growth and agricultural productivity has been a major global concern that requires urgent attention to avert future food scarcity [1]. Drought conditions occur due to the continuous increase in water scarcity required for plant use, most especially in the arid or semi-arid areas of the world [2]. Drought stress negatively affects the plant’s biochemical, physiological, and morphological processes, causing a reduction in plant yield, root architecture, photosynthetic rate, and nutrient assimilation [3]. The reduction in the yield of maize has been linked to limited water supply due to climate change [4,5]. Promisingly, exploring drought-tolerant plant-growth-promoting bacteria (PGPB) can be a suitable strategy for boosting plant adaptation and survival in low-water-regime areas [6]. Many studies have validated the potential of PGPB associated with the rhizosphere of tomato, wheat, maize, soybean, etc., as bioinoculants to enhance plant growth and ameliorate drought-induced stress on plants [5,7,8,9].

Plant-growth-promoting rhizobacteria (PGPR) are a group of bacteria found in the rhizosphere, on surfaces of the root, and around plant roots, which can indirectly or directly enhance the quality or yield of plants [10]. Plants’ reactions to abiotic or biotic stresses may be influenced by the close interface between rhizospheric microbes and the host plants [11,12]. The rhizosphere can be regarded as the narrow region in the soil environment, which can be directly influenced by root exudate secretions. Indirect enhancement in crop plant growth occurs when PGPR lessens or prevents adverse effects of phytopathogens, while the direct enhancement involves either inoculating plants with growth-promoting substances that are bacteria-synthesized or enabling specific plant nutrients’ uptake from the soil environment [13]. Reports indicated that plants affect native populations of soil microbes, with each plant species believed to establish a community of a particular microbial biomass [14,15,16].

Equally, root exudates alter microbial community structures in response to environmental stimuli or stress that can additionally influence microbial diversity. Recently, approachable techniques have been employed especially for root microbiome isolation and characterization, which expedites their recruitment to promote plant development and growth in stressed environments [17]. The microbes also assist in cycling nutrients by aiding nitrogen and phosphorus uptake while abetting the modification of root architecture under drought conditions [18]. Also, 1-aminocyclopropane-1-carboxylate (ACC) deaminase production, which regulates plants’ level of ethylene and inhibits leaf senescence under drought conditions, has been reported as one of the astonishing impacts of PGPR on crops [1,19]. Similarly, rhizobacteria can produce 1-aminocyclopropane-1-carboxylate (ACC) deaminase, lower the detrimental impact of ethylene, mitigate stress in plants, and enhance plant growth in drought stress conditions [20].

Drought is among the foremost stresses being confronted by a host of crop plants as a result of extreme scarcity of water and constitutes one of the key factors ascribed to substantial global crop production losses [21]. Many investigations in the past few decades have paid attention to transgenic approaches and molecular breeding for fortifying drought-resistant characteristics in varieties of crops [22,23]. As an alternative to the aforementioned approaches, the use of plant microbiomes has been dubbed as one of the finest approaches, considering its very low-cost input and ecofriendly nature, and is consequently amassing worldwide attention for drought stress management [24].

Frantic efforts are, therefore, being made to cultivate drought-tolerant plants via biotechnological approaches and not excluding breeding. Although these procedures may have high-cost implications, a few studies have explored using helpful microorganisms that already exist in the root microbiome to improve drought tolerance in a variety of crops [25,26,27]. However, the precise mechanisms supporting these are yet to be fully understood. Furthermore, it has been demonstrated that rhizobacteria-producing 1-aminocyclopropane-1-carboxylate (ACC) deaminase can enhance permeability by boosting soil aggregation and maintaining higher water potential around the roots, thus resulting in increased nutrient uptake, increased plant growth, and a defense against drought stress [15]. Authors hypothesized that the soils used in this study harbored notable PGPR capable of increasing drought tolerance in maize in water-scarcity-prone areas. Although studies on the PGPR-inducing drought stress in maize are evident in the literature, nevertheless, the use of *Aeromonas caviae* A1-2 and *Aeromonas veronii* C7_8 as bioinoculants to boost maize resilience to drought stress in South Africa is less studied. Hence, our findings provided new insights on the two potent rhizobacteria, *Aeromonas caviae* A1-2 and *A. veronii* C7_8, isolated from the maize rhizosphere, by assessing their plant-growth-promoting and drought-tolerant abilities with recommendations for future exploration in ameliorating effects of drought stress on plant development and productivity.

## 2. Materials and Methods

### 2.1. Study Site and Collection of Soil Samples

Soil samples were collected from the rhizosphere of two different locations where maize plants were cultivated within the Agricultural Farmland of North-West University, Molelwane, South Africa. The maize farm was located behind the Animal Health Centre, North-West University, South Africa, with sampling sites’ geographical coordinates (25°47′25.24056″ S 25°37′8.17464″ E) and (25°47′30.14056″ S 25°37′9.27464″ E). Healthy maize plants with intact root systems were uprooted and the soil samples attached to the roots were taken into sterilized plastic bags, labeled, and transported in an ice-packed box to the Microbial Biotechnology Research Laboratory, North-West University, in South Africa. Samples were stored at −20 °C before use and for a further analysis. 

### 2.2. Isolation of Culturable Rhizobacteria 

Rhizosphere soil was collected from the maize root environment at a depth of 2–4 cm using a sterile soil auger. A Radwag weighing machine was used to weigh about 1 g of soil, which was then positioned in a test tube holding 9 mL of sterile distilled water. The soil samples functioned as the source material for rhizobacteria isolation. Approximately 0.1 mL from the dilutions 10^−7^ and 10^−6^ was plated on sterilized Luria agar, nutrient agar, and tryptic agar plates, respectively, prepared according to the manufacturer’s instructions. Pure cultures were obtained from the mixed cultures by repeated streaking on nutrient agar following the method of Ahmad et al. [28]. The plates were incubated for 24 h at 30 °C. The obtained pure cultures were maintained on nutrient agar slants for further characterization. 

### 2.3. Morphological, Microscopic, and Biochemical Characterization of Rhizobacteria

Morphological characteristics of the isolated rhizobacteria were assayed on nutrient agar by utilizing the pure culture of the bacterial isolates that was incubated for 24 h at 28 ± 2 °C. Colonies’ morphology, including shape, size, pigmentation, and color, was properly documented after incubation, while Gram staining of the bacterial isolates was also carried out. Furthermore, the biochemical characterization tests of the bacterial isolates, such as Voges Proskauer, oxidase, nitrate reduction, catalase, indole, citrate, and starch hydrolysis, were carried out using the standardized procedure as described by Clarke and Cowan [29]. The carbohydrate utilization ability of the bacterial isolates was carried out using sugars such as galactose, maltose, xylose, lactose, fructose, glucose, and sucrose. 

### 2.4. Drought Tolerance Screening of the Rhizobacteria

The evaluation of the bacterial isolates’ drought tolerance was carried out in a nutrient-broth medium amended with varied PEG8000 concentrations (0, 5, 10, 15, 20, 25, and 30%). The PEG800 has a CAS number (25322683) and Catalog number (P0131). A single colony from a plate for each strain was inoculated into the broth medium and incubated for 48 h at 28 °C under continuous shaking at 200 rpm [2]. The estimation of bacterial growth was conducted by recording the cultures’ optical density with the use of Merck’s UV spectrophotometer at 600 nm. The best isolates were selected and subsequently examined for their capacity to promote plant growth under drought conditions. 

### 2.5. Screening of Rhizobacteria for Plant-Growth-Promoting Traits

#### 2.5.1. Ammonia Production

The methods described by Cappuccino and Sherman [30] were employed for the ammonia production screening of the bacterial isolates in peptone water. The pure cultures of each drought-tolerant bacterial strain were grown freshly and inoculated separately in test tubes containing 10 mL of peptone water and incubated at 28 ± 2 °C for 4 days on a rotary shaker (SI-600, LAB Companion (Republic of Korea (South Korea, Seoul))) at 120 rpm. Nessler’s reagent (0.5 mL) was added to each test tube and allowed to stay for 5 min for color development. A positive test for the production of ammonia was confirmed with the development of a yellow-to-orange coloration. The uninoculated test tube served as a control. The experiment was performed in triplicate. 

#### 2.5.2. Phosphate Solubilization

Phosphate solubilization screening was carried out by a spot inoculation of each bacteria isolate on Pikovskaya’s media [31]. The Pikovskaya agar was composed of 5 g of tricalcium phosphate (Ca_3_(PO_4_)_2_), 10 g of glucose (C_6_H_12_O_6_), 0.002 g of manganese sulfate (MnSO_4_·H_2_O), 0.2 g of sodium chloride (NaCl), 0.2 g of potassium chloride (KCl), 0.1 g of magnesium sulfate (MgSO_4_), 0.5 g of ammonium sulfate ((NH_4_)_2_SO_4_), 0.5 g of a yeast extract, 15 g of agar, and 1000 mL of sterile distilled water at pH 7.0. The media were sterilized and allowed to cool before pour-plating. The rhizobacteria were spot-inoculated at the center of the Petri plates and incubated at 28 ± 1 °C for 4–5 days. The emergence of a clear zone around the bacteria colony implied a positive result.

#### 2.5.3. Siderophore Production

The methods of Schwyn and Neilands [32] were employed to determine the siderophore production of each bacterial isolate in a chrome-azurol S (CAS) medium composed of 10 mL of an Fe(III) solution (27 mg of FeCl_3_·6H_2_O and 83.3 µL of concentrated HCl in 100 mL of ddH_2_O) along with 72.9 mg of hexadecyltrimethyl ammonium bromide (HDTMA). Each fresh bacterial culture 24 h old on nutrient agar was aseptically picked with a sterilized inoculating loop and streaked on the sterilized chrome-azurol S (CAS) medium and incubated for 48–72 h at 37 °C. The appearance of yellow/orange coloration around the colonies implied siderophore production.

#### 2.5.4. Nitrogen Fixation

The rhizobacteria nitrogen fixation characteristics were determined according to AlAli et al. [33] in Jensen’s medium composed of sucrose at 20 g; K_2_HPO_4_ at 1.0 g; MgSO_4_·7H_2_O at 0.5 g; NaCl at 0.5 g; FeSO_4_·7H_2_O at 0.1 g; CaCO_3_ at 2.0 g; agar at 15.0 g; Na_2_MoO_4_ at 0.005 g; and 1 liter of sterile distilled water. The medium was adjusted to pH 7.2 before sterilization at 121 °C for 15 min. Twenty-four-hour-old rhizobacterial cultures were inoculated into the Petri plates containing Jensen’s medium and incubated at 28 ± 2 °C for 7 days. The presence of bacterial colonies on the Petri dishes during the incubation implied the ability of nitrogen fixation by the bacterial cultures.

#### 2.5.5. Indole-Acetic Acid (IAA) Production

The qualitative IAA was determined according to the methods of Loper and Schroth [34] at different 5 mM (D)L-tryptophan concentrations of 0, 100, and 200 mg in LB broth. Pure bacterial isolates were inoculated in the broth and incubated in a shaking incubator at an ambient temperature (28 ± 2 °C) for 5 days. Bacterial growth suspensions were subjected to centrifugation at 3000 rev/min for 30 min. In total, 2 mL of the bacterial supernatant was mixed with 4 mL of a Salkowski reagent (prepared by mixing 49 mL of 35% perchloric acid in 1 mL of 0.5 M ferric chloride solution) and two drops of orthophosphoric acid. The development of a pink color indicated the production of IAA.

#### 2.5.6. Hydrogen Cyanide (HCN) Determination

A slight modification of Bakker and Schippers’ [35] procedure was employed for HCN production by each bacterial isolate. Glycine at 4.4 g was used to enrich the nutrient agar medium. Whatman No. 1 paper was immersed in 1% sodium carbonate, and 0.5% picric acid was placed on the lid of the Petri plates sealed with paraffin and incubated at 28 ± 1 °C for 5 days. The Petri plates were examined daily for color changes in the filter paper. A color change from yellow to brown indicated a positive result. The uninoculated tube served as a control. The experiment was carried out in triplicate.

#### 2.5.7. 1-Aminocyclopropane-1-carboxylate (ACC) Deaminase Activity

Qualitative ACC deaminase activity for the bacterial isolates was determined according to the method of Duan et al. [36]. Screening for ACC deaminase activity of drought-tolerant rhizobacterial isolates was conducted based on their ability to use ACC as a sole nitrogen source. All the bacterial isolates were grown in 5 mL of the LB medium incubated at 28 °C at 120 rpm for 4 days. The cells were harvested by centrifugation at 3000× *g* for 5 min and washed twice with sterile 0.1 M Tris-HCl (pH 7.5) and resuspended in 1 mL of 0.1 M Tris-HCl (pH 7.5) and spot-inoculated on Petri plates containing modified DF (Dworkin and Foster) salts’ minimal medium [37]. The medium composition includes glucose, 2.0 g; gluconic acid, 2.0 g; citric acid, 2.0 g; KH_2_PO_4_, 4.0 g; Na_2_HPO_4_, 6.0 g; MgSO_4_·7H_2_O, 0.2 g; a micronutrient solution (CaCl_2_, 200 mg; FeSO4·7H_2_O, 200 mg; H_3_BO_3_, 15 mg; ZnSO_4_·7H_2_O, 20 mg; Na_2_MoO_4_, 10 mg; KI, 10 mg; NaBr, 10 mg; MnCl_2_, 10 mg; COCl_2_, 5 mg; CuCl_2_, 5 mg; AlCl_3_, 2 mg; NiSO_4_, 2 mg; distilled water, 1000 mL), 10 mL; and distilled water, 990 mL, supplemented with 3 mM ACC as the sole nitrogen source. Plates contained only DF salts’ minimal medium without ACC as a negative control and with (NH4)_2_SO_4_ (0.2% *w*/*v*) as a positive control. The plates were incubated at 28 °C for 72 h. Growths of isolates on ACC-supplemented plates were compared to negative and positive controls and were selected based on growth by utilizing ACC as a nitrogen source. 

### 2.6. Tolerance of Bacterial Isolates to pH, Temperatures, Salt Concentrations, and Heavy Metals 

#### 2.6.1. Bacterial Tolerance to Different pH

The nutrient broth, Luria–Bertani Broth, tryptic soy broth, and Reasoner’s 2A broth were used for this test. The broth medium was prepared and sterilized at 121 °C for 15 min to test the tolerance of bacterial isolates to various pH levels. The pH of the medium was adjusted to acidic (pH 4) by adding diluted 1 M NaOH, neutral (pH 7), and alkaline (pH 10) by adding diluted 1 M HCl. Some 20 mL of each bacteria overnight culture was divided equally and then added to a sterile 10 mL LB broth and homogenized. The bacterial inoculation was performed in triplicate, and the inoculated Luria–Bertani broth was incubated at 30 °C for 24 h on a rotary shaker at 150 rpm. The optical density of the bacterial growth was taken at 600 nm using a UV spectrophotometer (Thermo Spectronic; Merck, Tokyo, Japan). 

#### 2.6.2. Tolerance of Bacterial Isolates to Different Temperatures

The nutrient broth, Luria–Bertani Broth, tryptic soy broth, and Reasoner’s 2A broth were used for the test. The broth medium was prepared and sterilized at 121 °C for 15 min. Some 25 mL of Luria–Bertani broth was gently vortexed after each bacterial strain was inoculated. The inoculated broth from each media was incubated at 25, 30, 35, and 40 °C. A spectrophotometer (Thermo Spectronic; Meck, Pretoria, South Africa) was used to measure the bacterial growth’s optical density at 600 nm after 24 h of incubation. 

#### 2.6.3. Bacterial Growth Response to Different Salt Concentrations

The nutrient broth, Luria–Bertani Broth, tryptic soy broth, and Reasoner’s 2A broth were used for the test. The broth medium was prepared and sterilized at 121 °C for 15 min to test the effect of bacterial strain tolerance to various salt concentrations (NaCl). While the control included no salt (0%), each broth from the mentioned media was produced following the manufacturer’s instructions and supplemented with salt concentrations at 1%, 3%, and 5%. The tubes containing 10 mL of each broth added with various salt concentrations were incubated at 30 °C on a shaking incubator for 72 h after inoculation with 100 µL of an overnight bacteria culture. A UV spectrophotometer (Thermo Spectronic; Merck) was used to detect the absorbance of the bacterial growth at 600 nm. 

#### 2.6.4. Bacterial Tolerance to Different Heavy Metals

The nutrient broth, Luria−Bertani broth, tryptic soy broth, and Reasoner’s 2A broth were prepared and sterilized at 121 °C for 15 min. The media were supplemented with 100 mg/L of various heavy metals, Pb^2+^, Cr_2_O_7_^2−^, and Cd^2^*^+^,* in test tubes to test for the tolerance of each bacterial isolate to various heavy metals. Test tubes containing precisely 10 mL of each broth medium containing a different heavy metal were filled, sterilized at 121 °C for 15 min, inoculated with 1 mL of each bacteria strain overnight culture, and then incubated for 72 h at 30 °C on a rotary shaker at 150 rpm. After incubation at 600 nm, the optical density of the bacterial growth was measured using a UV spectrophotometer (Thermo Spectronic; Merck). 

### 2.7. Extraction of DNA and Genotypic Identification

Fresh bacterial cultures on nutrient agar plates were used for the DNA extraction. The DNA of bacteria was extracted with the Zymo DNA extraction kit (Zymo Research, Irvine, CA, USA; Cat. No. D6005) as specified in the manufacturer’s instructions. A NanoDrop spectrophotometer was, thereafter, used to determine extracted DNA samples’ concentrations at a 260 nm wavelength; agarose gel electrophoresis was used to determine the presence of DNA. Also, 10 μL of Hyper Ladder™ 1 kb and 4 μL of bacterial DNA samples were run in 1.0% agarose concocted through the heating of 1 g of agarose powder in 100 mL of 1 × TAE at 65 °C that lasted 4 min, followed by the addition of 10 μL of ethidium bromide. The bands of DNA were observed in a gel Doc machine.

#### 2.7.1. Polymerase Chain Reaction (PCR) Amplification

Partial nucleotide sequences of 16S rRNA and PCR amplification were used to carry out genotypic identification. The bacteria 16S rRNA gene partial nucleotide sequences were obtained using the direct sequencing process of amplified PCR products. The bacteria 16S rRNA gene was amplified by PCR for bar-coded pyrosequencing. The full-length 16S rRNA gene sequences of all bacteria were amplified by utilizing universal forward, 27F (5′-AGAGTTTGATCCTGGCTCAG-3′) as well as the reverse, 1541R (5′-AAGGAGGTGATCCAGCCGCA-3′) primers [38]. A 25 μL reaction volume for an individual bacteria isolate was made up of 12.5 μL of OneTaq 2X MasterMix with the Standard Buffer, ~5 ng of genomic DNA, 1 μM for each primer, and 9.5 μL of nuclease-free water. A PCR thermal cycler machine (BIO-RAD Laboratories, Hercules, CA, USA, C1000 Touch) was used to carry out PCR amplification based on the corresponding program conditions: 95 °C for 2 min; 30 cycles at 95 °C for 30 s, 72 °C for 30 s, and 55 °C for 30 s, with a last extension of 72 °C for approximately 5 min. The standard procedure was used to perform quality control of PCR amplification and sequence preprocessing as well as raw data processing. Amplified DNA was checked by running DNA samples in 1% agarose gel as well as in a gel Doc machine (BIO-RAD Laboratories), as previously highlighted. DNA samples of each bacterial isolate of about 20 μL were forwarded for sequencing (INQABA, Pretoria, South Africa).

#### 2.7.2. Alignment Sequence and Construction of the Phylogenetic Tree

To determine sequence identities and similarities of bacterial isolates, the BLAST program of the nucleotide sequence on the National Centre for Biotechnology Information (NCBI) database was employed. Accession numbers were given to the identified bacteria after deposition on the GenBank on the NCBI web server. The sequences isolated were additionally exposed to the alignment of multiple sequences via ClustalX through the use of a Bio-Edit program. The Maximum Parsimony Analysis of taxa with the Kimura 2-parameter model was used to plot the phylogenetic tree using the MEGA-X program (Version 11) [39]. The bootstrap method was used to test the phylogeny. 

### 2.8. Experimental Set-up and Maize Growth Conditions

The experimental set-up for maize growth under a greenhouse was achieved using a three-factorial complete randomized design (CRD). The three factors were considered, which include *Aeromonas caviae* A1-2, *A. veronii* C7_8, and drought or water stress. With a shovel and plastic bags, soil samples were taken from topsoil (0–25 cm) near the North-West University garden on the Mahikeng campus. Between July 2021 and February 2022, soil samples were collected, homogenized, and sieved using a circular sieve with a 5.6 mm diameter. The sieved soil samples were sealed in autoclavable plastic bags and sterilized for 1 h at 121 °C (autoclave temperature). To ensure complete sterilization, this process was repeated three times. 

In this study, three drought or water regimes—100, 50, and 25% field capacity (FC)—were employed. The FC calculation has been included in the Appendix A. *A. caviae* A1-2 and *A. veronii* C7_8 were grown in broth cultures and harvested according to the method of Prakamhang et al. [40] with little modifications. In flasks containing 1000 mL of nutrient broth, fresh bacterial cultures were cultivated for six days at 180 rpm in a shaker incubator (FMH200 Instruments manufactured by FireChief, located in East Sussex, United Kingdom). After incubation, the broth culture was centrifuged at 1000× *g* for 10 min, followed by two-step washing with a sterile 0.85% saline solution. The initial optical density (OD) of the bacterial strains was adjusted to 1.3 OD, and the bacterial titer was 30 × 10^9^ (colony-forming unit) mL^−1^. The experiment was carried out in plastic pots (30 cm in diameter and 29 cm in height), each with 6 kg of sterilized soil. A total of 192 pots containing two maize plants were produced by setting up 24 treatments with four replicates per treatment.

The PAN 413 cultivar of maize seeds was cleaned in sterile distilled water, surface-sterilized in 75% ethanol, and then thoroughly rinsed with 1% sodium hypochlorite to eliminate all traces of the chemicals. Then, approximately 240 surface-sterilized maize seeds were dispensed in 20 mL broth cultures of both *A. caviae* A1-2 and *A. veronii* C7_8 adjusted to 1.3 OD of an equal volume for 1 h at 28 °C. An equal volume (10 mL) of each bacterial strain was combined for seed bacterialization. The flasks containing seeds were shaken for 24 h at 180 rpm at 28.2 °C to properly combine the seeds and inocula. In a sterilized laminar flow cabinet (Filta Matix Laminar Flow Cabinet), the liquid suspension was decanted and air-dried on sterile aluminum foil sheets before seeding. Approximately 6 kg of sterilized soil was placed in each container in the greenhouse. Ten maize seeds were planted per pot. The seeds and seedlings in the container were inoculated with 10 mL bacterial cultures separately at growth intervals over time. The seeds dispensed inside sterile distilled water without bacterial inoculation served as the control. The seed inoculation was carried out, but direct inoculation onto the seed inside the pots and seedling roots appeared after 7 days of growth. The seeds’ viability was validated at the Agricultural Research Council, Pretoria, South Africa (supplier of the maize seeds). 

Two weeks following emergence, the seedlings were reduced to one per container and harvested at 45 days of maturity. The above- and below-ground yield parameters were measured. For about three weeks after germination, all the water treatments (100, 50, and 25% FC) were fully watered to the field capacity at 72-hour intervals. Thereafter, drought or water stress was initiated in the 50 and 25% FC treatments, and they were watered every 48 h with 335 mL and 170 mL of water, respectively. However, the 100% FC treatments (the watering control) continued to be watered with 670 mL of water. The pots were watered every 72 hours until the experiment’s end, increasing drought stress three weeks after it was first introduced [41].

### Below-Ground–Above-Ground/Morphological Parameters 

The chlorophyll content, plant height, leaf count, plant diameter, fresh total plant mass (FTPM), fresh root mass (FRM), fresh shoot mass (FSM), dry total mass (DTM), dry root mass (DRM), and dry shoot mass (DSM) were measured and determined after harvesting the mature maize plants. The roots of the maize plant were thoroughly cleaned, and then the plant’s fresh weights were ascertained right away by weighing the entire plant on a scale (Wagi Elek’roniczne, city of Bieniądzice, located in Wieluń, Poland). The shoot was then cut away from the roots. Using a measuring tape, the shoot height and root length were determined; the number of completely grown branches, leaves, and lateral roots (excluding the root) were then counted to determine the branch, leaf, and lateral root number. After being separated, the shoots and roots were oven-dried at 70 °C for 48 h and then weighed to determine their dry weights. A chlorophyll content meter (CCM-200 plus) was used to measure the chlorophyll content of the youngest completely grown leaf at the distal end of the maize plant.

### 2.9. Parameters Measured 

#### 2.9.1. Leaf Relative Water Content 

The leaf relative water content was determined according to the methods described by Ortiz et al. [42] with little modifications, and the “youngest fully developed leaves of each plant” were used. Fresh leaf samples were weighed (fresh weight—FW), placed in test tubes saturated with water, and kept at 4 °C for 48 h. Thereafter, the leaf samples were weighed again to obtain the turgid weight (TW) and oven-dried at 60 °C for 24 h to obtain the dry weight (DW). The leaf relative water content was calculated as follows:$$\mathrm{Leaf}\;\mathrm{relative}\;\mathrm{water}\;\mathrm{content}\;\left(\%\right)=\frac{\left(\mathrm{FW}-\mathrm{DW}\right)}{\left(\mathrm{TW}-\mathrm{DW}\right)}\times 100$$

#### 2.9.2. Electrolyte Leakage 

The Ortiz et al. [42] approach was used to assess electrolyte leakage. From each of the four replicates for each treatment, a young leaf of nearly the same size from the distal end of the maize stalk was taken and thoroughly washed with deionized water to remove any electrolytes that could have adhered to the leaf surface. After being incubated for 24 h at 28 ± 2 °C on a rotational shaker, leaf samples were deposited in sealed test tubes. The electrical conductivity (Lt) of the sample solution was then calculated using a Taiwanese-made PL-700AL conductivity meter. Before determining the final electrical conductivity (L_0_), samples were autoclaved at 120 °C for 20 min and then allowed to cool to 25 °C. The following formula was used to determine the electrolyte leakage:$$\mathrm{Electrolyte}\;\mathrm{leakage}\;(\%)\;\frac{Lt}{L0}\;\times \;100\%$$

#### 2.9.3. Soluble Sugar Content in Maize Leaves 

One hundred milligrams (100 mg) of maize leaves was homogenized in 4 mL of 80% ethanol and centrifuged for 10 min at 5000 rpm. In total, 1 mL of the supernatant was mixed with 5 mL of concentrated sulphuric acid (H_2_SO_4_) and 1 mL of 5% phenol. Samples were vigorously agitated and allowed to cool for 30 min, and absorbance was measured at 485 nm using a spectrophotometer (Thermo Spectronic, South Africa). The concentration of soluble sugar was calculated from a standard curve “established with a reference glucose solution”, as described by Zarik et al. [43] with little modifications. Briefly, stock solutions of glucose were prepared by weighing 10, 20, 30, 40, 50, 60, and 70 mg of D(+)-glucose in 10 mL of sterile distilled water. Two millileters of each stock solution was pipetted into three cleaned test tubes. In total, 1 mL of 5% phenol was added, and 5 mL of concentrated H_2_SO_4_ was quickly added. The tubes were allowed to stand for 10 min, then shaken and placed in a water bath for 10 min at 51–57 °C. Tubes were allowed to cool for 30 min, and absorbance was measured at 485 nm using a spectrophotometer and a standard curve was plotted from the values obtained.

#### 2.9.4. Proline Content in Maize Leaves 

Proline in maize leaves was extracted using the Bates et al. [44] technique with only minor modifications. Briefly, 1.25 g of ninhydrin was dissolved in 20 mL of 6 M phosphoric acid and 30 mL of glacial acetic acid by heating on a hot plate with agitation. The solution was allowed to cool and kept at 4 °C and the solution became stable after 24 h. Approximately 1 g of a fresh maize leaf sample was ground in 10 mL of 3% aqueous sulfur-salicylic acid and centrifuged at 10,000× *g* for 10 min. In total, 2 mL of the supernatant was reacted with 2 mL of glacial acetic acid and 2 mL of an acid–ninhydrin solution in 45 mL falcon tubes at 100 °C in a water bath for 60 min, and the reaction was stopped in an ice box. Four millileters of toluene was added to extract the mixture and agitated vigorously for 15–20 s in a shaker incubator at 250 rpm. The mixture was kept in the dark for 30 min, the “chromophore containing toluene was aspirated from the aqueous phase, and the absorbance was read at 520 nm using toluene for a blank. The concentration of proline was estimated from a standard curve” established with a reference proline solution. Briefly, a 1 mg/mL stock solution of proline was prepared by weighing 10 mg of proline (DL-Proline, Hefei, China) in 10 mL of sterile water. A stock solution of 0, 50, 100, 150, 200, 250, and 300 μL was pipetted into seven tubes containing 300, 250, 200, 150, 100, 50, and 0 μL of sterile water, respectively. The mixtures were then reacted with 2 mL of glacial acetic acid and 2 mL of the acid–ninhydrin solution in 45 mL falcon tubes at 100 °C in a water bath for 60 min, and the reaction was stopped in an ice box. The mixture was vigorously agitated using a vortex (Vortex Genie, New York, NY, USA) after adding 4 mL of toluene. The mixture was kept in the dark for 30 min, and the absorbance of the proline-containing upper layer was read at 520 nm using toluene for a blank, and the proline standard curve was plotted from the absorbance values.

### 2.10. Statistical Analyses 

All the study’s data were collected in three replicates at various time intervals, and they were all processed using SPSS version 26 and Microsoft Excel software version 26. To evaluate the effects of each treatment at various drought levels, the data were subjected to a general linear model analysis of variance (ANOVA). Differences between means (post hoc analysis) were determined by Duncan’s multiple range test (DMRT) [45], and significant differences were set at *p* ≤ 0.05. 

## 2.11. Data Availability Statement

The sequence data of the bacteria strains A5-1, A1-2, B8-3, B12-4, B9-5, B15-6, C6-7, C7_8, C1-9, A9-10, and A10-11 were deposited in the NCBI GenBank with accession numbers ON745408, ON745409, ON745410, ON745411, ON745412, ON745413, ON745414, ON745415, ON745416, ON745417, and ON745418, respectively.

## 3. Results

### 3.1. Characterization of the Rhizobacteria Strains

The microscopic appearance and biochemical characteristics of the rhizobacterial isolates are presented (Appendix A and Appendix A). The microscopic observations of the bacterial isolates revealed different structural appearances under the light microscope (Appendix A). Also, the biochemical tests showed different results (Appendix A). All the bacterial strains showed 100% similarity compared with the NCBI database, except for strain A1-2, which showed 99.83% similarity. 

The biochemical characteristics and Gram reaction test showed that A5-1 was Gram-positive short-rod motile, while B8-3, B12-4, B9-5, B15-6, and C1-9 were Gram-positive rod motile; similarly, bacterial strain C6-7 was Gram-positive motile and Bacilli, respectively. Furthermore, bacteria strains A9-10 and A10-11 were Gram-positive nonmotile cocci and rods, respectively. All the bacterial isolates were positive for catalase, in addition to being oxidase-positive for the entire strains except bacterial B8-3 and B15-6, which were negative for catalase, and bacteria strains B12-4, C6-7, C1-9, and A9-10, which were negative for oxidase. All the isolates tested positive for maltose, glucose, and sucrose. Isolate B12-4 was not determined for nitrate reduction, while isolate A9-10 was not determined for xylose. The phylogeny of the *Aeromonas* strains A1-2 and C7_8 with high drought tolerance is presented in Figure 1 and Figure 2.

### 3.2. Selection of Drought-Tolerant Rhizobacteria Strains

The identifiable rhizobacteria strains with distinctive drought tolerance levels in a broth medium amended with PEG-8000 are presented in Table 1. The rhizobacterial strains C7_8 and A1-2 were selected based on their high response drought tolerance values of 0.77 and 0.71 in a medium supplemented with 30% PEG-8000 compared with the control that showed low drought tolerance. Similarly, the rhizobacterial strain A5-1 showed a negligible value of 0.12 at 30% of PEG amendments.

### 3.3. Plant-Growth-Promoting Abilities of Rhizobacterial Strains and Phylogeny

Table 2 shows the plant-growth-promoting traits of the screened rhizobacteria. It was observed that all the rhizobacterial strains exhibited positive reactions to nitrogen fixation, siderophore, ammonia, IAA, phosphate solubilization, and ACC deaminase activity. All the bacterial strains tested negative for hydrogen cyanide production. Similarly, seven out of eleven bacterial isolates exhibited a relatively high level of ammonia production. Meanwhile, the bacteria strain *B. cereus* B8-3 exhibited moderate ammonia production. Based on the additional evaluation, the rhizospheric bacteria that belong to *Bacillus licheniformis*, *Aeromonas caviae*, *Priestia flexa*, and *Aeromonas veronii* exhibited high positive reactions to siderophore production compared to other strains. *Aeromonas caviae* A1-2, *Priestia flexa* C6-7, and *Aeromonas veronii* C7_8 showed a high level of indole-acetic acid (IAA) compared to other strains. 

### 3.4. Rhizobacteria Tolerance to Drought Stress under Different Growth Conditions

The rhizobacteria tolerance to drought under different growth conditions is presented in Table 3, Table 4, Table 5 and Table 6. The rhizobacteria *Aeromonas caviae* (A1-2) and *Aeromonas veronii* (C7_8) were subjected to salt stress at 0% (control), 1%, 3%, and 5% concentrations in different nutrient media, nutrient broth (NB), Luria−Bertani broth (LBB), tryptic soy broth (TSB), and Reasoner’s 2A broth (R2AB). Both rhizobacteria strain A1-2 and C7_8 displayed various levels of salt tolerance at the 5% salt concentration compared with the control (Table 3). 

The growth of isolated rhizobacterial strains A1-2 and C7_8 exposed to pH-stimulated stress is presented in Table 4. The bacteria showed varied tolerance to drought at different pH measurements of 4, 7, and 10 in the growth medium. Furthermore, the rhizobacterial isolates showed tolerance by varying the temperatures of the medium at 25, 30, 35, and 40 °C (Table 5). High bacterial tolerance to drought was recorded at 35 °C compared with the control. This study, in addition, affirms the tolerance of isolated rhizobacteria *Aeromonas caviae* (A1-2) and *Aeromonas veronii* (C7_8) to heavy metals, Pb^2+^, Cr_2_O_7_^2−^, and Cd^2+^ (Table 6). Similarly, rhizobacterium A1-2 exhibited high activity of 0.76, 0.63, and 0.50 under Pb^2+^, Cr_2_O_7_^2-^, and Cd^2+^ stress, while the rhizobacterium C7_8 metal tolerance level was 0.74, 0.61, and 0.49 under Pb^2+^, Cr_2_O_7_^2−^, and Cd^2+^ stress, respectively. The use of nutrient broth shows considerable and potential support for the growth of the bacteria under different growth conditions. Therefore, NB is considered the best medium for the selected strains under different conditions. 

### 3.5. Rhizobacteria Improve the Growth of Maize Plants Exposed to Varying Levels of Drought Stress in a Greenhouse 

The untreated and treated maize with rhizobacteria, *A. caviae* (A1-2), and *A. veronii* (C7_8) against drought stress in the greenhouse is presented in Table 7. The response of the maize plants against drought stress was measured concerning the plant height (cm), number of leaves, leaf area (cm^2^), stem girth (cm), and chlorophyll content. It was observed that a combination of both bacteria offered the best stress tolerance level for the maize against drought stress, as maize treated with both bacteria showed better performance at 100, 50, and 25 field capacity (FC). The combined treatment of bacteria strains (A1-2+C7_8) displayed a high plant height of 9.33 cm, 8.77 cm, and 8.38 cm compared to the control with the low plant height of 7.27 cm, 6.08 cm, and 6.10 cm at 100, 50, and 25 FC, respectively (Table 7). Similar observations were recorded for stem girth, number of leaves, leaf area, and chlorophyll content. The combined treatment of strains A1-2+C7_8 had the highest leaf numbers of 7.08, 6.50, and 6.50; stem girth of 4.58, 4.53, and 4.18 cm; and chlorophyll content of 14.13, 14.64, and 13.64 at 100, 50, and 25 FC, respectively, compared with the control. In terms of leaf area, the highest values of 85.13, 72.37, and 73.80 cm^2^ at 100, 50, and 25 FC were recorded compared with the control. 

The response of the plants to varied drought stress was shown every week (Figure 3). The combined treatment of selected rhizobacteria strains, A1-2+C7_8, showed a better drought tolerance response compared to the control. All the treatments showed similar pattern increments in terms of plant height at 100 FC (Figure 3). However, the combined treatments showed an upward considerable drought tolerance response at day 25 concerning plant length (Figure 3), stem girth (Figure 3), leaf number (Figure 3), chlorophyll content (Figure 3), and leaf area (Figure 3) at 100, 50, and 25 FC. Hence, the response of the maize plants to drought stress can be arranged in decreasing order as A1-2+C7_8 > A1-2 > C7_8 > control, corresponding to extremely effective > moderately effective > slightly effective, respectively. 

### 3.6. Rhizobacteria Improving the Biomass of Maize Exposed to Drought Stress in the Greenhouse

The response of maize plants to drought stress enhanced by rhizobacteria treatment was evaluated in the greenhouse. The plants treated with 20 mL each of the rhizobacteria, *A. caviae* (A1-2), and *A. veronii* (C7_8) showed improved biomass compared to the control (Table 8). In detail, the combined treatments of A1-2+C7_8 had high fresh total plant mass of 19.41 g, 16.68 g, and 10.29 g, while maize treated with *A. caviae* (A1-2) had fresh total plant biomass of 18.70 g, 13.45 g, and 10.52 g at WC of 100, 50, and 25, respectively, compared with the control. A fresh root mass of 8.42 g, 4.35 g, and 3.91 g of treated maize with strains A1-2+C7_8 at water capacities of 100, 50, and 25 was recorded. The maize treated with *A. caviae* (A1-2) showed higher performance than those treated with *A. veronii* (C7_8) compared with the control. Based on the biomass, the response of the maize to drought stress can be arranged in a decreasing order as follows: A1-2+C7_8> A1-2 > C7_8 > control, corresponding to extremely effective > moderately effective > slightly effective, respectively. 

### 3.7. Rhizobacteria Improve the Morphophysiological Attributes of Maize Plants Exposed to Drought Stress in the Greenhouse

The rhizobacterial treatment of maize plants to withstand varying levels of drought stress was also established on some morphophysiological characteristics of the plant concerning the % leaf relative water content (Figure 4), % electrolyte leakages (Figure 5), sugar content (mg/mL) (Figure 6), and proline content (µmol/g of dry weight) (Figure 7). It was observed that the co-inoculation of *A. caviae* A1-2 and *A. veronii* C7_8 enhanced the relative water content (RWC) of the plant leaves both under drought stress at 50 and 25% water capacity (WC) and under unstressed conditions at 100 WC (Figure 4). The singular inoculation of the bacterial strains exerted a slight enhancement in the RWC of the plants at 100, 50, and 25 WC compared with the control, which showed a decrease with an increase in drought stress. The leaves of the uninoculated maize plants significantly increased the electrolyte leakages from 50 to 25 WC, while the singular and combined treatment showed a reduction in the electrolyte leakages (Figure 5). The soluble sugar content in the maize plants both under stress and under unstressed conditions is presented in Figure 6. It was observed that the combined rhizobacterial strains’ treatment enhanced the sugar content of the leaves compared with the control (*p* ≤ 0.05). The combined maize treatment with A1-2+C7_8 had higher glucose content than that treated with either *A. caviae* (A1-2) or *A. veronii* (C7_8) singularly. In Figure 7, the rhizobacterial treatment significantly enhanced the proline content in maize both under drought-stressed (50 and 25 WC) and unstressed conditions (100 WC) compared to the control. 

## 4. Discussion 

The rhizosphere represents discreet regions of plant−soil−microbe interactions with the host-plant−root interface [46]. As such, the rhizosphere is home to many plant-growth-promoting rhizobacteria (PGPR), with literature evidence related to their beneficial effects on plants and how they can be better exploited for sustainable agriculture and crop production [16,47,48]. The plant-growth-enhancing efficiency of rhizobacteria is considerably dependent on their capability to subsist and create an effective colonization of the plant root [49,50]. In addition, efficient plant root colonization by PGPR is crucial in promoting growth regardless of their mode of action [51]. Therefore, screening of drought tolerance PGPR should be carried out to evaluate their potential and drought stress reduction in plants. Limited studies have identified PGPR *Aeromonas* species from some plants in arid and semi-arid regions [52,53]. Therefore, there is a need to study further the potential impact of the mono- and co-inoculation of the rhizobacteria strains on maize in the greenhouse and field experimental conditions. Many identifiable PGPR belonging to the genera *Bacillus*, *Pantoea*, *Gluconacetobacer*, *Pseudomonas*, and *Paenibacillus* with plant-growth-promoting attributes have been reported [53,54,55,56]. Hence, this study affirms the biostimulatory potentials of rhizobacteria for maize improvement under drought stress in a greenhouse. 

The isolation and characterization of drought-tolerant PGPR associated with maize were carried out in this study. Two of the identifiable bacterial isolates exhibited high drought tolerance, coupled with the PGP properties. The rhizobacterial strains, *Aeromonas caviae* A1-2 and *A. veronii* C7_8, at 30% PEG amendment induced high drought tolerance, thus alleviating the effect of drought stress on the inoculated maize and enhancing their growth. The phylogeny revealed that the two bacterial strains belong to the genera *Aeromonas*, which are most prevalent in agrarian soils. A greater number of *Aeromonas* species have been extensively investigated for plant improvement [57,58]. According to Arshad et al. [59], PGPR that possess ACC deaminase activity aid crop plants to withstand drought stress by lowering the ethylene (stress hormone) production level through the hydrolysis of ACC into ammonia and α-ketobutyrate. These ACC-deaminase-producing bacteria have already been documented to enhance plant growth, particularly in stress conditions, such as heavy metals, flooding, high salt, and drought [21].

In a study carried out by Nawaz et al. [52], the authors characterized PGPR, *Aeromonas* strains SAL-17 and SAL-21, isolated from two wheat genotypes and soybean to produce acyl-homoserine lactone (AHL), which assisted plants to withstand stress conditions. Sarkar et al. [60] reported ACC deaminase activity of halotolerant *Enterobacter* sp. to enhance plant growth. Furthermore, the high ACC-deaminase-producing potential of *Aeromonas* strains A1-2 and C7_8 corroborates the findings of Cruz et al. [61], who reported *Aeromonas* sp. in other crops by confirming their potentialities in ameliorating drought stress for plant survival. The ACC deaminase activity of the *Aeromonas* strains A1-2 and C7_8 may be associated with their viability/survival in the PEG-amended medium and genetic makeup to assist crops’ survival under stress [62]. Drought tolerance genes aid bacteria in withstanding adverse conditions and colonizing endorhizosphere regions through the linkage of fibrillar material, which fosters bacterial attachment to the surfaces while protecting plants [63]. 

The PGPR are well known to support plant growth via symbiotic association. Typically, rhizobacteria inoculation can cause an improvement in plant growth through diverse mechanisms, such as the production of siderophores, IAA, ACC deaminase activity, nitrogen fixation, and other growth-promoting features [64]. The production of phytohormones can alter plant root architecture to improve water and nutrient uptake and retention. The ability of PGPR to ameliorate abiotic stressors in the environment is just one of the advantages underlining various metabolic and physiological inductions in the host plants. In addition to the in vitro growth assessment, greenhouse studies have shown that growth traits exhibited by the rhizobacteria strains A1-2 and C7_8 contributed significantly to the seed germination index compared to the control. 

The present studies focus on the enhancement in the symbiotic plant−soil−microbe mechanisms for the improvement in maize production. We employed rhizobacteria strains A1-2 and C7_8 as PGPR and demonstrated how their inoculation into maize roots supports the growth and development of maize under the greenhouse experiment. This mechanism relies on the exchange of materials between the plant and the microbes encompassing a better decomposition of organic, nutrient release, and cycling, as well as the induction of systemic resistance against biotic and abiotic stresses [65]. Niu et al. [62] and Danish et al. [6] had demonstrated earlier the induction of drought tolerance in plants using rhizosphere microbes. The observed increase in the maize seed germination rate was consistent with earlier reports, which support the use of microbial agents, singularly or combined with other agents, to mediate physiological changes and improve crop growth [66,67].

Consequently, plant inoculation with drought-tolerant rhizobacteria with manifold growth-promoting tendencies can enhance the effectiveness of the bacteria inoculants, especially in semi-arid or arid regions [62]. The rhizobacteria strains, A1-2 and C7_8, exhibited varied drought tolerance at varied salinity, pH, temperature, and heavy metal stress, which stimulated plant growth when they co-existed to ameliorate effects of the stressors and corroborated the findings of Abedinzadeh et al. [68], who affirmed the potential effects of abiotic stressors on plant growth, through enzyme and protein synthesis. The results from this study are similar to the findings of Fasusi et al. [69], who affirmed the plant-growth-promoting tendencies of PGPR isolated from food plants in South Africa. 

It is well known that cereals and legumes like soybean and maize are popular food crops of economic importance and are of immense value for human diets and feed consumption. Their cultivation, however, can easily be affected by different factors ranging from soil infertility, pest disease, and abiotic stressors. Hence, the need to improve their production to meet the demand of the increasing world population remains paramount [70]. The use of many synthetic agrochemicals in modern crop production is on the rise, with concurrent increases in cost and residual harmful effects on the environment and resultant health risks to the consumers along the food value chain [71,72]. Many PGPR have been reported to stimulate plant growth and survival against different environmental stresses [73,74,75]. 

Among several recent studies on rhizobacteria as biostimulants, Kunwar et al. [76] reported the use of rhizobacteria *Pseudomonas* species to stimulate maize plants through nutrient bioavailability and acquisition. Albareda et al. [77] suggested the use of peat as a carrier of Rhizobia. Kunwar et al. [76] reported on *P. syringe* + peat and *P. putida* + clay; they demonstrated that the solid biostimulants improved the plant yield in terms of height, stem diameter, and leaf number and area. In another study, Siddiq et al. [78] showed that newly isolated *B. thuringiensis* from Pakistan significantly increased bean growth and yield. Scott et al. [79] established the enhancement in corn growth, nutrients, and chlorophyll content using rhizobacterium *Azospirillum brasilense* HM053.

Our study established the importance of rhizobacteria strains A1-2 and C7_8 in the improvement in maize plants against drought stress, which corroborates previous findings of Vacheron et al. [80], who reported the effect of biostimulatory agents on maize yield. Diarrassouba et al. [81] employed *P. fluorescens* as a bioinoculant to improve the growth of two varieties of *Zea mays* L. in soils under low water capacity. Likewise, Noumavo et al. [82] also reported the induction of metabolic and fungicidal potentials of maize using some locally isolated rhizobacteria strains. In a more recent study, Amogou et al. [83] established the application of some locally isolated rhizobacteria from Benin to enhance maize growth. Notably, Mejri et al. [84] formulated a deleterious rhizobacterium, *Pseudomonas trivialis* X33d, as a biocontrol agent against the Durum wheat pathogen *Bromus diandrus*. Additionally, Agbodjato et al. [85] reported synergistic interactions between rhizobacteria and chitosan for the in vitro acclimatization of maize under germination and nutrient uptake under greenhouse conditions. The improvement in the maize growth and yield subjected to drought stress under the greenhouse experiment agrees with the findings of Janardan et al. [86] and Kumar et al. [87], who demonstrated the effect of some rhizobacteria in seed germination and growth enhancement in chickpea (*Cicer arietinum* L.) and wheat (*Triticum aestivum* L.) under greenhouse conditions. Therefore, biostimulants using PGPR can be considered over synthetic chemicals, which can pose residual effects and health hazards.

Under harsh conditions, the host plant tends to respond to drought-induced stress by producing osmolytes to help the plant sustain osmotic balancing in econiches [88]. The ability of plants to produce osmolytes, such as glycine betaine, total soluble sugars, proline, and total protein under drought stress, can help mitigate the effects of oxidative stress on plants. Osmolyte production in plants can be enhanced by bioinoculation with the copious agriculturally important rhizobacteria with diverse growth-promoting characteristics [89]. The maize treated with the combined *A. caviae* A1-2 + *A. veronii* C7_8 enhanced the sugar contents of the leaves compared with the control. The combined maize treatment with A1-2+C7_8 also exhibited a higher glucose content than the maize treated with either *A. caviae* A1-2 or *A. veronii* C7_8 singularly under drought stress. The massive production of osmolytes in maize inoculated with *A. caviae* A1-2 + *A. veronii* C7_8 can signally help in sustaining cell membrane integrity and water status and the prevention of protein degradation under drought stress. The results obtained in this study established the bacterial inoculant co-existing with maize and production of enhanced soluble sugar under drought stress, which indicated the drought stress amelioration potential of *A. caviae* A1-2 and *A. veronii* C7_8 on the maize plants grown in water-deficit areas.

The seed inoculation enhanced the crop yield, and this can be linked to the ability of the bacterial strains to adapt to and survive harsh conditions. Our current studies align with the literature stating crops within naturally drought-stressed conditions are expedient cradles of drought-tolerant bacteria with plant-growth-enhancing prospects. The inferences from this study support the hypothesis that plant PGPR can exhibit important input in the drought locale plant adaptation, such as maize. Hence, we propose that drought-tolerant rhizobacteria, *A. caviae* A1-2, and *A. veronii* C7_8, can further be explored as bioinoculants in managing abiotic drought stress in maize and other staple food crops.

## 5. Conclusions

This study affirmed that drought tolerance and notable PGPR strains A1-2 and *A. veronii* C7_8 can alleviate drought stress in maize plants upon inoculation, either singularly or combined. Notably, the results of this study established that the inoculation of identifiable rhizobacteria *Aeromonas* strains A1-2 and C7_8 had significant effects on maize growth under drought stress. The bioinoculation supported the physiological and biochemical functions of the maize plants. It also culminated in improving both the below-ground–above-ground and/or morphological components of maize plants compared to the noninoculated plants. The combined effects of *A. caviae* A1-2 and *A. veronii* C7_8 were observed to significantly enhance maize yields, such as plant height, leaf number, stem girth, leaf area and chlorophyll content, and plant shoot and maize biomass, such as fresh total plant mass, fresh root mass, fresh shoot mass, dry total plant mass, dry root mass, and dry shoot mass under drought stress. The results obtained can provide more insights into the impacts of the rhizobacteria on maize growth and provide theoretical background for the real-life application of these strains in field experimental conditions. This biorational strategy can help resolve or abate food insecurity problems in most developing countries in the semi-arid and arid regions of the world. Nevertheless, an additional investigation of these bacterial strains can help unravel their effectiveness as suitable candidates in the formulation of biofertilizers. Furthermore, omics studies can promisingly help to unravel the identity and actual mechanisms of *Aeromonas* species in a natural plant environment for future exploration in enhancing crop yield, promoting plant growth under drought stress, and ensuring food security.

## Figures and Tables

**Figure 1 plants-13-01298-f001:**
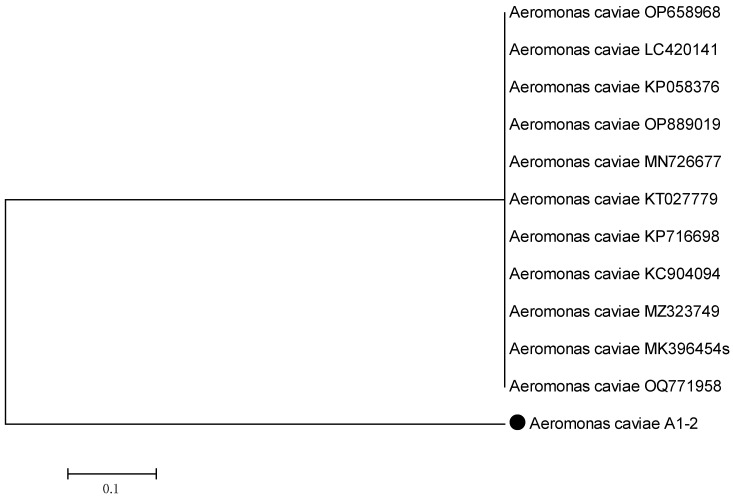
Evolutionary relationships of the taxa tree based on partial 16S rRNA sequences using maximum likelihood. The tree shows correlations between the rhizospheric *A. caviae* A1-2 and their closely associated strains from the NCBI GenBank. The boxes represent the sequences generated in this study.

**Figure 2 plants-13-01298-f002:**
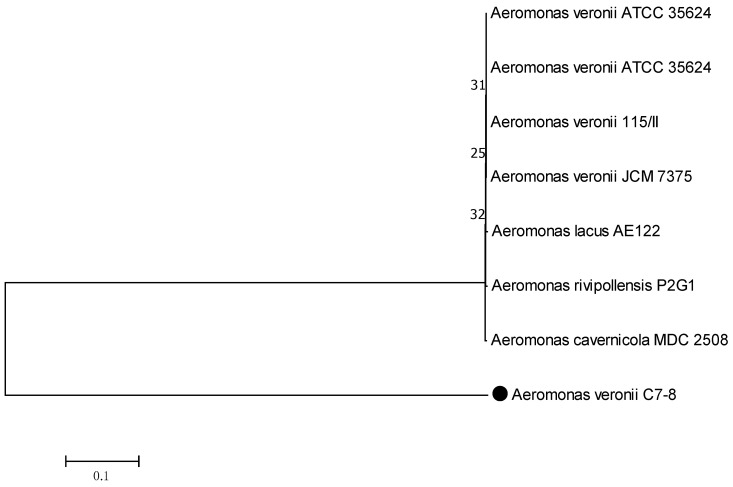
Evolutionary relationships of the taxa tree based on partial 16S rRNA sequences utilizing maximum likelihood that shows correlations between the rhizospheric *A. veronii* C7_8 and their closely associated strains from the NCBI GenBank. The boxes represent the sequences generated in this study.

**Figure 3 plants-13-01298-f003:**
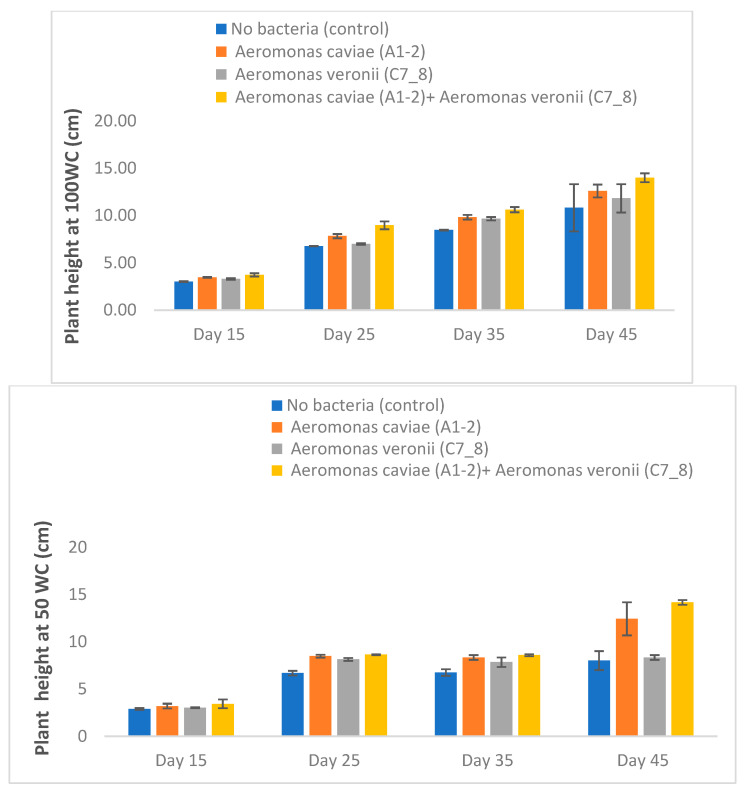
The growth response of maize plants exposed to rhizobacterial treatments at varying periods of drought stress. No bacteria = Uninoculated plants (control), Treatment 1 = Maize plants with rhizobacterium *Aeromonas caviae* A1_2, Treatment 2 = Maize plants with rhizobacterium *Aeromonas veronii* C7_8, Treatment 1 and 2 = Maize plants with rhizobacteria *Aeromonas caviae* A1_2 and C7_8.

**Figure 4 plants-13-01298-f004:**
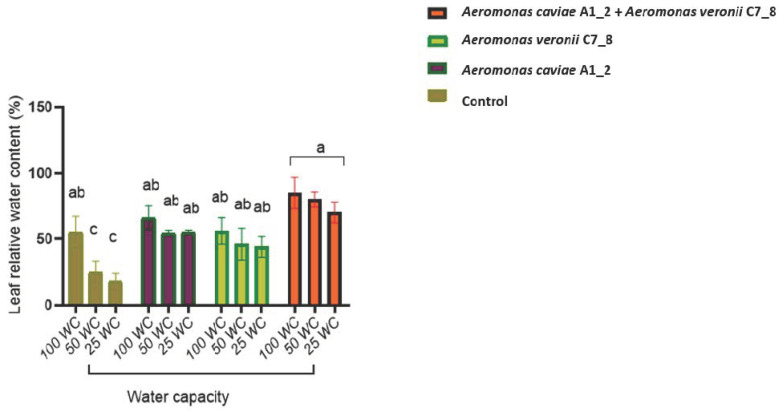
Relative water content of maize leaves inoculated and noninoculated with rhizobacteria, *A. caviae* (A1-2), and *A. veronii* (C7_8) under varying levels of drought stress. Bars are means of three replicates ± SD. Control—noninoculated treatment, and combined strains A1-2+C7_8.

**Figure 5 plants-13-01298-f005:**
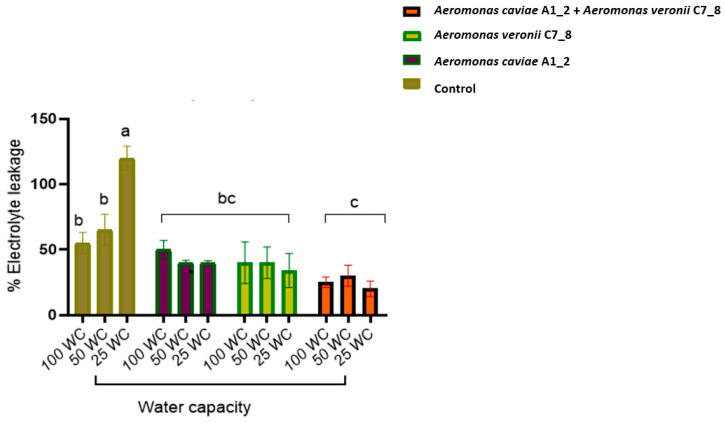
Electrolyte leakage of maize leaves inoculated and noninoculated with rhizobacteria *A. caviae* (A1-2) and *A. veronii* (C7_8) under varying levels of drought stress. Bars are means of three replicates ± SD. Control—noninoculated treatment, and combined strains A1-2+C7_8.

**Figure 6 plants-13-01298-f006:**
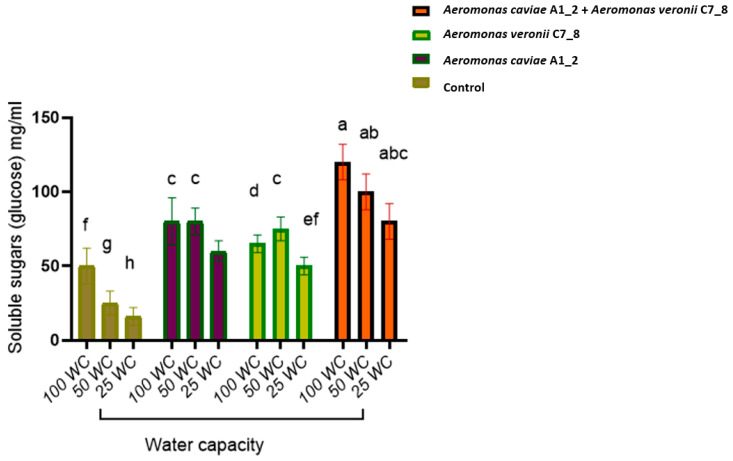
Soluble sugar accumulation in maize of maize leaves inoculated and noninoculated with rhizobacteria, *A. caviae* (A1-2), and *A. veronii* (C7_8) under varying levels of drought stress. Bars are means of three replicates ± SD. Control—noninoculated treatment.

**Figure 7 plants-13-01298-f007:**
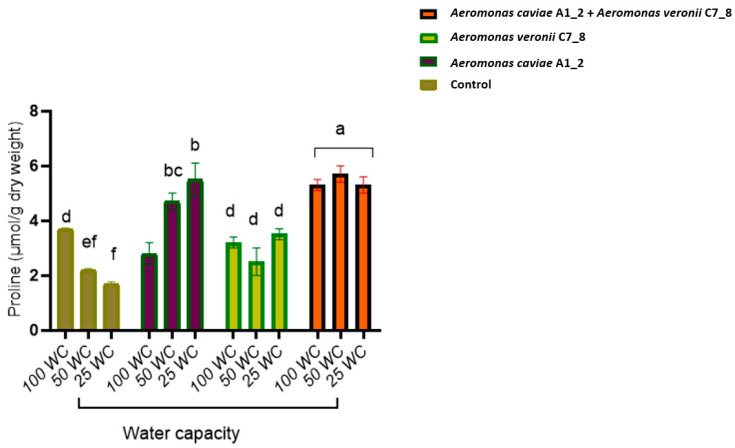
Proline accumulation in maize of maize leaves inoculated and noninoculated with rhizobacteria, *A. caviae* (A1-2), and *A. veronii* (C7_8) under varying levels of drought stress. Bars are means of three replicates ± SD. Control—noninoculated treatment.

**Table 1 plants-13-01298-t001:** Response of rhizobacterial strains to drought stress amended with various concentrations of PEG-8000.

Treatment	Control	5%	10%	15%	20%	25%	30%
A5-1	0.35 ± 0.17fg	0.18 ± 0.01c	0.16 ± 0.01d	0.16 ± 0.00f	0.16 ± 0.00d	0.13 ± 0.00f	0.12 ± 0.00c
A1-2	1.01 ± 0.07b	0.67 ± 0.23a	0.80 ± 0.15a	0.89 ± 0.03a	0.87 ± 0.03a	0.81 ± 0.01a	0.71 ± 0.01a
B8-3	0.73 ± 0.03bc	0.38 ± 0.11ab	0.41 ± 0.02bc	0.39 ± 0.00c	0.37 ± 0.02b	0.30 ± 0.00c	0.26 ± 0.03bc
B12-4	0.65 ± 0.08cde	0.40 ± 0.11abc	0.41 ± 0.01bc	0.41 ± 0.01c	0.41 ± 0.00b	0.35 ± 0.01b	0.30 ± 0.00bc
B9-5	0.21 ± 0.03g	0.24 ± 0.05c	0.21 ± 0.02cd	0.20 ± 0.00ef	0.20 ± 0.00d	0.18 ± 0.01e	0.15 ± 0.01c
B15-6	0.43 ± 0.08egf	0.27 ± 0.05c	0.31 ± 0.02cbd	0.31 ± 0.00d	0.28 ± 0.00c	0.22 ± 0.00d	0.20 ± 0.00c
C6-7	0.84 ± 0.06bc	0.40 ± 0.11abc	0.45 ± 0.04b	0.41 ± 0.00c	0.40 ± 0.0.00b	0.32 ± 0.01bc	0.29 ± 0.00bc
C7_8	1.31 ± 0.13a	0.53 ± 0.20ab	0.73 ± 0.15a	0.84 ± 0.03b	0.84 ± 0.03a	0.81 ± 0.04a	0.77 ± 0.03a
C1-9	0.35 ± 0.01fg	0.33 ± 0.08bc	0.31 ± 0.02bcd	0.21 ± 0.00e	0.20 ± 0.00d	0.19 ± 0.00de	0.14 ± 0.01c
A9-10	0.23 ± 0.03g	0.24 ± 0.04c	0.17 ± 0.00d	0.17 ± 0.01ef	0.16 ± 0.00d	0.17 ± 0.01e	0.16 ± 0.01c
A10-11	0.55 ± 0.05edf	0.25 ± 0.04c	0.20 ± 0.01cd	0.18 ± 0.01ef	0.17 ± 0.01d	0.17 ± 0.01e	0.15 ± 0.01c
LSD	0.22	0.23	0.20	0.04	0.04	0.04	0.24

The values are represented as the mean ± standard error with the lowercase letters indicating significant differences down the column.

**Table 2 plants-13-01298-t002:** Plant-growth-promoting characteristics of rhizobacterial strains.

Rhizobacterial Isolates	Plant-Growth-Promoting Traits						
	Nitrogen Fixation	HCN Production	Siderophore Production	Ammonia Production	IAA Production	Phosphate Solubilization	ACC Deaminase
*B. licheniformis* A5-1	+	–	++	+++	+	+	+
*A. caviae* A1-2	+	–	++	++++	++	+	+
*B. cereus* B8-3	+	–	+	+++	+	+	+
*P. flexa* B12-4	+	–	+	++	+	+	+
*B. licheniformis* B9-5	+	–	+	++	+	+	+
*B. simplex* B15-6	+	–	+	++	+	+	+
*P. flexa* C6-7	+	–	++	+++	++	+	+
*A. veronii* C7_8	+	–	++	++++	++	+	+
*P. aryabhattai* C1-9	+	–	+	++	+	+	+
*B. halotolerans* A9-10	+	–	+	+++	+	+	+
*P.endophytica* A10-11	+	–	+	+++	+	+	+

Key: + = positive; − = negative.

**Table 3 plants-13-01298-t003:** Growth of isolated rhizobacterial strains exposed to salt-stimulated stress.

Rhizobacteria	Media	0% Salt	1% Salt	3% Salt	5% Salt
*Aeromonas caviae* (A1-2)	NB	1.64 ± 0.18 ^a^	1.42 ± 0.56 ^a^	1.37 ± 0.60 ^a^	0.55 ± 0.02 ^b^
	LB	1.62 ± 0.18 ^a^	1.34 ± 0.56 ^ab^	0.96 ± 0.56 ^b^	0.50 ± 0.02 ^c^
	TSB	1.47 ± 0.12 ^a^	1.28 ± 0.56 ^ab^	0.90 ± 0.56 ^b^	0.45 ± 0.02 ^c^
	R2AB	1.37 ± 0.12 ^a^	1.22 ± 0.51 ^a^	1.18 ± 0.56 ^a^	0.40 ± 0.02 ^b^
*Aeromonas veronii* (C7_8)	NB	1.61 ± 0.18 ^a^	1.42 ± 0.54 ^a^	0.70 ± 0.543 ^bc^	0.45 ± 0.12 ^c^
	LB	1.60 ± 0.13 ^a^	1.35 ± 0.61 ^ab^	1.00 ± 0.61 ^abc^	0.41 ± 0.12 ^c^
	TSB	1.52 ± 0.25 ^a^	1.28 ± 0.55 ^ab^	0.96 ± 0.56 ^abc^	0.37 ± 0.12 ^c^
	R2AB	1.4224 ± 0.3154 ^a^	1.22 ± 0.60 ^ab^	0.93 ± 0.57 ^abc^	0.38 ± 0.18 ^c^

The mean value ± SD with the same superscript letter across each row was not significantly different according to the Duncan multiple range test (DMRT) at *p* ≤ 0.05. Number of replicates (n) = 4. NB = nutrient broth, LB = Luria−Bertani broth, TSB = tryptic soy broth, R2AB = Reasoner’s 2A broth.

**Table 4 plants-13-01298-t004:** Growth of isolated rhizobacterial strains exposed to pH-stimulated stress.

Rhizobacteria	Media	pH4	pH7	pH10
*Aeromonas caviae* (A1-2)	NB	0.97 ± 0.05 ^ab^	1.11 ± 0.02 ^a^	0.56 ± 0.32 ^c^
	LB	0.88 ± 0.05 ^b^	1.03 ± 0.27 ^a^	0.50 ± 0.29 ^c^
	TSB	0.79 ± 0.04 ^b^	0.96 ± 0.30 ^a^	0.45 ± 0.26 ^c^
	R2AB	0.71 ± 0.04 ^b^	1.03 ± 0.56 ^a^	0.27 ± 0.01 ^c^
*Aeromonas veronii* (C7_8)	NB	0.91 ± 0.09 ^ab^	1.53 ± 0.60 ^a^	0.40 ± 0.01 ^c^
	LB	0.82 ± 0.07 ^ab^	1.11 ± 0.56 ^a^	0.36 ± 0.01 ^c^
	TSB	0.74 ± 0.07 ^ab^	1.36 ± 0.59 ^a^	0.32 ± 0.01 ^c^
	R2AB	0.66 ± 0.06 ^ab^	1.29 ± 0.60 ^a^	0.30 ± 0.02 ^c^

The mean value ±SD with the same superscript letter across each row is not significantly different according to the Duncan multiple range test (DMRT) at *p* ≤ 0.05. Number of replicates (n) = 4. NB = nutrient broth, LB = Luria−Bertani Broth, TSB = tryptic soy broth, R2AB = Reasoner’s 2A broth.

**Table 5 plants-13-01298-t005:** Growth of isolated rhizobacterial strains exposed to temperature-stimulated stress.

Rhizobacteria	Media	25 °C	30 °C	35 °C	40 °C
*Aeromonas caviae* (A1-2)	NB	0.93 ± 0.03 ^a^	1.20 ± 0.01 ^a^	1.72 ± 0.01 ^a^	0.83 ± 0.03 ^a^
	LB	0.87 ± 0.07 ^a^	1.08 ± 0.01 ^a^	1.59 ± 0.17 ^a^	0.75 ± 0.032 ^b^
	TSB	0.99 ± 0.42 ^a^	0.97 ± 0.01 ^a^	1.21 ± 0.48 ^a^	0.67 ± 0.02 ^b^
	R2AB	0.88 ± 0.01 ^b^	1.01 ± 0.59 ^a^	1.32 ± 0.58 ^a^	0.61 ± 0.02 ^b^
*Aeromonas veronii* (C7_8)	NB	0.97 ± 0.01 ^a^	1.09 ± 0.01 ^a^	1.64 ± 0.05 ^a^	0.82 ± 0.04 ^a^
	LB	0.87 ± 0.01 ^b^	0.98 ± 0.01 ^b^	1.55 ± 0.01 ^a^	0.74 ± 0.04 ^c^
	TSB	0.88 ± 0.01 ^b^	1.02 ± 0.40 ^a^	1.40 ± 0.01 ^a^	0.66 ± 0.03 ^c^
	R2AB	1.04 ± 0.57 ^a^	1.26 ± 0.01 ^a^	0.79 ± 0.01 ^b^	0.60 ± 0.03 ^b^

The mean value ± SD with the same superscript letter across each row was not significantly different according to the Duncan multiple range test (DMRT) at *p* ≤ 0.05. Number of replicates (n) = 4. NB = nutrient broth, LB = Luria−Bertani broth, TSB = tryptic soy broth, R2AB = Reasoner’s 2A broth.

**Table 6 plants-13-01298-t006:** Growth of isolated rhizobacterial strains exposed to heavy-metal-stimulated stress.

Rhizobacteria	Media	0 (Control)	Pb^2+^	Cr_2_O_7_^2-^	Cd^2+^
*Aeromonas caviae* (A1-2)	NB	1.28 ± 0.54 ^a^	0.76 ± 0.08 ^ab^	0.63 ± 0.10 ^b^	0.50 ± 0.09 ^b^
	LB	1.56 ± 0.22 ^a^	0.68 ± 0.08 ^b^	0.56 ± 0.09 ^bc^	0.44 ± 0.08 ^c^
	TSB	1.51 ± 0.13 ^a^	0.61 ± 0.07 ^b^	0.51 ± 0.08 ^c^	0.40 ± 0.07 ^c^
	R2AB	1.08 ± 0.36 ^a^	0.55 ± 0.06 ^b^	0.46 ± 0.07 ^b^	0.36 ± 0.06 ^c^
*Aeromonas veronii* (C7_8)	NB	1.26 ± 0.53 ^a^	0.74 ± 0.08 ^b^	0.61 ± 0.10 ^b^	0.49 ± 0.09 ^c^
	LB	1.26 ± 0.35 ^a^	0.67 ± 0.01 ^b^	0.55 ± 0.09 ^b^	0.44 ± 0.08 ^c^
	TSB	1.15 ± 0.34 ^a^	0.60 ± 0.07 ^b^	0.50 ± 0.08 ^b^	0.40 ± 0.07 ^c^
	R2AB	1.0787 ± 0.3217 ^a^	0.54 ± 0.06 ^b^	0.45 ± 0.07 ^c^	0.36 ± 0.06 ^c^

The mean value ± SD with the same superscript letter across each row was not significantly different according to the Duncan multiple range test (DMRT) at *p* ≤ 0.05. Number of replicates (n) = 4. NB = nutrient broth, LB = Luria−Bertani broth, TSB = tryptic soy broth, R2AB = Reasoner’s 2A broth.

**Table 7 plants-13-01298-t007:** Effects of rhizobacterial inoculation on the growth of maize plants exposed to varying periods of drought stress.

Water Regime	Growth Parameters	Control	A1-2	C7_8	A1-2+C7_8
100 WC	Plant height (cm)	7.27 ± 0.95 ^c^	8.43 ± 0.43 ^ab^	7.95 ± 0.50 ^b^	9.33 ± 0.34 ^a^
	Number of leaves	5.67 ± 0.29 ^d^	6.67 ± 0.14 ^b^	6.50 ± 0.29 ^c^	7.08 ± 0.29 ^a^
	Stem girth (cm)	3.08 ± 0.21 ^c^	4.19 ± 0.31 ^b^	3.73 ± 0.18 ^bc^	4.58 ± 0.27 ^a^
	Chlorophyll content	9.12 ± 0.71 ^bcd^	11.78 ± 2.22 ^b^	10.13 ± 0.92 ^bc^	14.13 ± 2.74 ^a^
	Leaf area (cm^2^)	52.60 ± 5.35 ^c^	79.67 ± 4.30 ^ab^	63.81 ± 2.89 ^abc^	85.13 ± 14.13 ^a^
50 WC	Plant height (cm)	6.08 ± 0.45 ^d^	8.11 ± 0.63 ^ab^	6.83 ± 0.27 ^c^	8.71 ± 0.23 ^a^
	Number of leaves	5.92 ± 0.14 ^a^	6.58 ± 0.43 ^a^	6.08 ± 0.14 ^a^	6.50 ± 0.29 ^a^
	Stem girth (cm)	3.06 ± 0.23 ^b^	3.75 ± 0.41 ^ab^	3.45 ± 0.20 ^ab^	4.53 ± 0.32 ^a^
	Chlorophyll content	8.44 ± 0.52 ^c^	10.69 ± 1.49 ^b^	9.57 ± 0.89 ^b^	14.64 ± 3.56 ^a^
	Leaf area (cm^2^)	46.88 ± 1.76 ^c^	66.57 ± 8.72 ^ab^	44.44 ± 0.52 ^abc^	72.37 ± 0.73 ^a^
25 WC	Plant height (cm)	6.10 ± 0.30 ^d^	7.68 ± 0.75 ^ab^	6.62 ± 0.29 ^c^	8.38 ± 0.53 ^a^
	Number of leaves	6.00 ± 0.29 ^a^	6.42 ± 0.39 ^a^	6.08 ± 0.14 ^a^	6.50 ± 0.00 ^a^
	Stem girth (cm)	3.17 ± 0.11 ^b^	3.73 ± 0.36 ^ab^	3.43 ± 0.13 ^b^	4.18 ± 0.34 ^a^
	Chlorophyll content	9.09 ± 0.92 ^c^	12.93 ± 1.98 ^ab^	10.92 ± 1.57 ^abc^	13.64 ± 1.50 ^a^
	Leaf area (cm^2^)	42.67 ± 11.19 ^c^	55.94 ± 2.03 ^a^	47.78 ± 1.29 ^bc^	73.80 ± 0.82 ^a^

Values are means of three replicates ± SD; values across each row having the same superscript letters are not significant according to the Duncan multiple range test (DMRT) at *p* ≤ 0.05. TO = uninoculated plants (Control), A1-2 = plants whose roots were inoculated with rhizobacterium *Aeromonas caviae* (Treatment 1), C7_8 = plants whose roots were inoculated with rhizobacterium *Aeromonas veronii* (Treatment 2), A1-2+C7_8 = plants whose roots were co-inoculated with rhizobacteria *Aeromonas caviae* and *Aeromonas veronii* (Treatment 3).

**Table 8 plants-13-01298-t008:** Effects of rhizobacterial inoculation on the biomass of maize exposed to drought stress.

Water Capacity	Parameters (g)	Control	A1-2	C7_8	A1-2+C7_8
100	Fresh Total Plant Mass	10.32 ± 0.22 ^c^	18.70 ± 4.36 ^ab^	11.85 ± 1.01 ^b^	19.41 ± 4.01 ^a^
	Fresh Root Mass	3.59 ± 1.69 ^d^	6.28 ± 1.75 ^ab^	3.52 ± 0.45 ^c^	8.42 ± 2.31 ^a^
	Fresh Shoot Mass	6.49 ± 1.58 ^d^	10.06 ± 2.14 ^b^	8.28 ± 0.98 ^c^	13.19 ± 2.53 ^a^
	Dry Total Plant Mass	1.45 ± 0.03 ^d^	3.22 ± 0.68 ^ab^	1.81 ± 0.23 ^cd^	3.45 ± 0.62 ^a^
	Dry Root Mass	0.53 ± 0.17 ^cd^	1.24 ± 0.35 ^ab^	0.58 ± 0.06 ^cd^	1.35 ± 0.34 ^a^
	Dry Shoot Mass	1.03 ± 0.09 ^bcd^	1.87 ± 0.34 ^ab^	1.21 ± 0.19 ^abc^	2.12 ± 0.28 ^a^
50	Fresh Total Plant Mass	5.75 ± 2.12 ^cd^	13.45 ± 3.58 ^b^	7.53 ± 1.31 ^bc^	16.68 ± 0.90 ^a^
	Fresh Root Mass	1.70 ± 0.85 ^c^	4.59 ± 0.98 ^a^	3.93 ± 0.58 ^b^	4.35 ± 0.48 ^a^
	Fresh Shoot Mass	4.03 ± 1.27 ^cd^	8.72 ± 2.52 ^ab^	3.58 ± 1.86 ^d^	12.31 ± 1.36 ^a^
	Dry Total Plant Mass	1.07 ± 0.06 ^bcd^	2.15 ± 0.71 ^ab^	1.33 ± 0.14 ^bc^	2.80 ± 0.24 ^a^
	Dry Root Mass	0.37 ± 0.05 ^d^	0.91 ± 0.36 ^ab^	0.47 ± 0.08 ^bc^	0.96 ± 0.03 ^a^
	Dry Shoot Mass	0.64 ± 0.03 ^d^	1.43 ± 0.49 ^ab^	0.78 ± 0.08 ^abc^	1.77 ± 0.22 ^a^
25	Fresh Total Plant Mass	5.12 ± 0.44 ^cd^	10.52 ± 3.86 ^ab^	6.69 ± 1.20 ^bc^	10.29 ± 2.85 ^a^
	Fresh Root Mass	1.49 ± 0.21 ^c^	2.03 ± 0.33 ^ab^	1.75 ± 1.08 ^abc^	3.91 ± 0.70 ^a^
	Fresh Shoot Mass	2.94 ± 0.07 ^bcd^	6.61 ± 3.04 ^b^	5.14 ± 1.36 ^bc^	8.54 ± 1.78 ^a^
	Dry Total Plant Mass	0.91 ± 0.07 ^d^	1.72 ± 0.49 ^ab^	1.21 ± 0.13 ^c^	1.86 ± 0.56 ^a^
	Dry Root Mass	0.39 ± 0.05 ^bcd^	0.50 ± 0.13 ^b^	0.43 ± 0.03 ^ac^	0.74 ± 0.21 ^a^
	Dry Shoot Mass	0.51 ± 0.01 ^cd^	0.98 ± 0.27 ^ab^	0.78 ± 0.13 ^c^	1.27 ± 0.34 ^a^

Values are means of three replicates ± SD; values across each row having the same superscript letters are not significant according to the Duncan multiple range test (DMRT) at *p* ≤ 0.05. Uninoculated plants (Control), A1-2 = plants whose roots were inoculated with rhizobacterium *A. caviae* (Treatment 1), C7_8 = plants whose roots were inoculated with rhizobacterium *A. veronii* (Treatment 2), A1-2+C7_8 = plants whose roots were co-inoculated with *A. caviae* and *A. veronii* (Treatment 3).

## Data Availability

The original contributions presented in the study are included in the article/Appendix A, further inquiries can be directed to the corresponding author/s.

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
