# Peer review of "Isolation and Characterization of Plant-Growth-Promoting, Drought-Tolerant Rhizobacteria for Improved Maize Productivity"

_plants, 2024, doi:10.3390/plants13101298_

Round 1

Reviewer 1 Report

Comments and Suggestions for Authors

The topic Genomic Characterization of Plant Growth Promoting-Drought Tolerant Rhizobacteria Isolated from Maize Plants is important topic also the manuscript is well written and all section are connected with eachother. But after reading this article i suggest that add some new literature specially in the introduction section, avoid literature older than 2015. Also the hypothesis at the end of introduction is missing. Please add it and also improve the quality of figure 4. Once you addressed these comments then the article will be acceptabale for publication.

Author Response

AUTHORS’ RESPONSE TO REVIEWER 1

Comments and Suggestions for Authors

The topic Genomic Characterization of Plant Growth Promoting-Drought Tolerant Rhizobacteria Isolated from Maize Plants is important topic also the manuscript is well written and all section are connected with each other.

RESPONSE: Authors’ appreciate reviewer 1 for the brilliant review and commendation on our manuscript.

But after reading this article i suggest that add some new literature especially in the introduction section, avoid literature older than 2015.

RESPONSE: New literature have been added to the introduction section as suggested (Line 31-43; 86-95). Generally, the introduction section has been thoroughly revised.

Also the hypothesis at the end of introduction is missing. Please add it

RESPONSE: The hypothesis of the study has been revised appropriately (Line 86-91)

and also improve the quality of figure 4.

RESPONSE: This is now Figure 3. The quality has been improved and revised appropriately with more clarity (Line 596)

Once you addressed these comments then the article will be acceptabale for publication.

RESPONSE: Authors’ appreciate Reviewer 1 for this recommendation of acceptance after revision.

Reviewer 2 Report

Comments and Suggestions for Authors

Article title:

“Genomic Characterization of Plant Growth Promoting-Drought Tolerant Rhizobacteria Isolated from Maize Plants".

               The work is unique, really interesting, and valid for publication after some modification points:   Major comments:

-          During the isolation, three media (Luria agar, Nutrient agar, and Tryptic agar), were used. Then, one media (NA) was used in evaluations of growth regulators and drought tolerance. Why?

-          Added 2.9. Tolerance of bacterial isolates to pH, temperatures, salt concentrations, and heavy metals before 2.8. Extraction of DNA and Genotypic Identification. Rearrange again in all the sections of the manuscript.

-          What about "The growth media used" in Bacterial tolerance to different pH (Line 207), different temperatures (Line 216), different salt concentrations (Line 224), and different heavy metals (Line 234)?

-          Why was the anion radical not proven in the experiment? Line 236.

-          Line 249: and sterilised for 15 minutes at 121°C. How it’s a soil (it's better to sterilize to 1 h).

-          How to calculate the FC (100, 50, 25%)?

-          Added 2.12. Physicochemical Parameter Analyses of Soil Samples in the first MM section.

-          Added 3.1. Characterization of the Rhizobacteria Strains in supplementary file (Figure 1, Table 1).

-          It would have been better to perform tolerance to drought stress under different growth conditions tests for all tested isolates, as is the case with growth regulators (PGP)

    Minor comments: Abstract:

- Line 10: Bioinucula change to bioinocula

- Line 15 and 16: Change Bacillus to B.

- Line 21:  combined.An increase change to combined. An increased

Introduction: Acceptable

Materials and methods:

-          Line 110: add CAS No of Lot No of the chemical used of PEG8000.

-          Line 111: (0%, 5%, 10%, 15%, 20%, 25%, 30%). Change to (0, 5, 10, 15, 20, 25 and 30%).

-          Line 112: delete distance " continuous    shaking ".

-          Line 121-122: A rocking abbreviat was used for incubation at 28±2°C for 4 days. was poured into each of the tubes and allow to stay for 5 minutes. (Rewrite again).

-          Line 140: MgSO47H2O 0.5 g; NaCI 0.5 g; change to MgSO4 7H2O 0.5 g; NaCl 0.5 g;

-          Line 140: Complete the component of the media used such as pH, and sterilized conditions. Etc

-          Line 146: tryptophan (L or D)

-          Line 211: please l, ?????

-          Line 244: Added the design of the experiment in Experimental Set-up and Maize Growth Conditions

-          Line 251: A. caviae, write completely.

-          Line 262: twenty 24 treatments???

-          Line 262 and 263: Rewrite again

-          Line 301: RWC measured, what is the time?

-          Line 303: added the number for the formula.

-          Line 306: From each of the eight replicates for each treatment????   8 or 4 replicates.

-          Line 315: added the number for the formula.

-          Line 317: Soluble sugars in soybean leaves were extracted. Oh, Soybean or maize?????

-          Line 317 -328: Rewrite again

-          Line 330: Proline in maize leaves measured, what the time?

-          Line 342: DELETE ?????.

Results

-          Table 2: LSD for 1%, 5% or….? Added

-          Below Table 2 " Table 1. The value were represented as the mean of 3 replicates (n=3) ± SE. Values with different lowercase letters are different significantly." ????

-          The phylogeny of the Aeromonas strains A1-2 and C7_8 with high drought tolerance is presented in Figures 2, 3, Added to 3.1. Characterization of the Rhizobacteria Strains.

-            Lines 447- 456: Check for Figure 6 of 4.

Discussion

-          Line 119: Daei et al. (2009), Hart et al. (2012) and 119 Sánchez-Matamoros et al. (2018) have.., change to style of the journal.

-          Line 173: can exhibit cigent input ?????  

Conclusions

-       Line 178-182: delete.

-       Lin 197: food insecurity problem. Change to the food insecurity problems

-       Line 202: promote plant growth under drought stress and ensure food security change to promoting plant growth under drought stress and ensuring food security

Comments on the Quality of English Language

The overall writing is good. However, a number of problems were detected throughout the MS, including grammar errors. please check the MS to make sure there is no language problem

Author Response

AUTHORS’ RESPONSE TO REVIEWER 2

Comments and Suggestions for Authors

Article title:

“Genomic Characterization of Plant Growth Promoting-Drought Tolerant Rhizobacteria Isolated from Maize Plants".

The work is unique, really interesting, and valid for publication after some modification points:   Major comments:

RESPONSE: Authors’ appreciate reviewer 2 for the brilliant review and finding our manuscript to be unique, interesting and valid for publication.

-          During the isolation, three media (Luria agar, Nutrient agar, and Tryptic agar), were used. Then, one media (NA) was used in evaluations of growth regulators and drought tolerance. Why?

RESPONSE: The NA was used because of high microbial counts recorded compared to other media. Hence, it was used for other tests (Line 116-117).

-          Added 2.9. Tolerance of bacterial isolates to pH, temperatures, salt concentrations, and heavy metals before 2.8. Extraction of DNA and Genotypic Identification. Rearrange again in all the sections of the manuscript.

RESPONSE: This has been rearranged and revised appropriately (Line 234-278).

-          What about "The growth media used" in Bacterial tolerance to different pH (Line 207), different temperatures (Line 216), different salt concentrations (Line 224), and different heavy metals (Line 234)?

RESPONSE: The growth media used has been revised appropriately (Line 236-237; 249-250; 258-259, 270-271).

-          Why was the anion radical not proven in the experiment? Line 236.

RESPONSE: This has been revised appropriately (Line 272).

-          Line 249: and sterilised for 15 minutes at 121°C. How it’s a soil (it's better to sterilize to 1 h).

RESPONSE: This has been revised appropriately (Line 330).

-          How to calculate the FC (100, 50, 25%)?

RESPONSE: This has been added to the Supplementary file document (Line 333).

-          Added 2.12. Physicochemical Parameter Analyses of Soil Samples in the first MM section.

RESPONSE: This section has been expunged from the body of the manuscript.

-          Added 3.1. Characterization of the Rhizobacteria Strains in supplementary file (Figure 1, Table 1).

RESPONSE: This has been revised and taken to supplementary file (465).

-          It would have been better to perform tolerance to drought stress under different growth conditions tests for all tested isolates, as is the case with growth regulators (PGP)

RESPONSE: Authors appreciate the reviewer 2 suggestion, which is remarkable and contributively. However, the scope of this research was as presented. Nevertheless, the suggestions shall be considered in our future studies.

    Minor comments: Abstract:

- Line 10: Bioinucula change to bioinocula

RESPONSE: Revised appropriately (Line 10)

- Line 15 and 16: Change Bacillus to B.

RESPONSE: Revised appropriately (Lines 15-17)

- Line 21:  combined.An increase change to combined. An increased

RESPONSE: Revised appropriately (Lines 22)

 Introduction: Acceptable

RESPONSE: Authors appreciate Reviewer 2 for accepting our Introduction section as presented. However, authors have revised this section based on other reviewers’ comments.

 Materials and methods:

-          Line 110: add CAS No of Lot No of the chemical used of PEG8000.

RESPONSE: Revised appropriately (Line 138)

-          Line 111: (0%, 5%, 10%, 15%, 20%, 25%, 30%). Change to (0, 5, 10, 15, 20, 25 and 30%).

RESPONSE: Revised appropriately (Line 137)

-          Line 112: delete distance " continuous    shaking ".

RESPONSE: Revised appropriately (Line 140)

-          Line 121-122: A rocking abbreviat was used for incubation at 28±2°C for 4 days. was poured into each of the tubes and allow to stay for 5 minutes. (Rewrite again).

RESPONSE: The statement has been revised appropriately (Line 152-154)

-          Line 140: MgSO47H2O 0.5 g; NaCI 0.5 g; change to MgSO4 7H2O 0.5 g; NaCl 0.5 g;

RESPONSE: Changed and revised appropriately (Line 185-186)

-          Line 140: Complete the component of the media used such as pH, and sterilized conditions. Etc

RESPONSE: Revised appropriately (Line 185-189)

-          Line 146: tryptophan (L or D)

RESPONSE: Revised appropriately (Line 195)

-          Line 211: please l, ?????

RESPONSE: Revised appropriately (Line 240)

-          Line 244: Added the design of the experiment in Experimental Set-up and Maize Growth Conditions

RESPONSE: Revised appropriately (Line 323-324)

-          Line 251: A. caviae, write completely.

RESPONSE: This sentence has been expunged and revised appropriately from Line 331)

-          Line 262: twenty 24 treatments???

RESPONSE: Revised appropriately (Line 342-343)

-          Line 262 and 263: Rewrite again

RESPONSE: The sentence has been rewritten and revised appropriately (Line 342-343)

-          Line 301: RWC measured, what is the time?

RESPONSE: The sentence has been revised appropriately (Line 387-393)

-          Line 303: added the number for the formula.

RESPONSE: The information of the formula has been well presented and revised appropriately (Line 395)

-          Line 306: From each of the eight replicates for each treatment????   8 or 4 replicates.

RESPONSE: Four replicates have been revised appropriately (Line 401)

-          Line 315: added the number for the formula.

RESPONSE: Revised appropriately (Line 409)

-          Line 317: Soluble sugars in soybean leaves were extracted. Oh, Soybean or maize?????

RESPONSE: Revised appropriately (Line 411)

-          Line 317 -328: Rewrite again

RESPONSE: This sub-section has been re-written and revised appropriately (Line 410-424)

-          Line 330: Proline in maize leaves measured, what the time?

RESPONSE: This sub-section has been revised appropriately (Line 426-448)

-          Line 342: DELETE ?????.

RESPONSE: This has been expunged from the manuscript.

 Results

 -          Table 2: LSD for 1%, 5% or….? Added

RESPONSE: Now Table 1. It has been revised appropriately (Line 488)

-          Below Table 2 " Table 1. The value were represented as the mean of 3 replicates (n=3) ± SE. Values with different lowercase letters are different significantly." ????

RESPONSE: Revised appropriately (Line 489)

-          The phylogeny of the Aeromonas strains A1-2 and C7_8 with high drought tolerance is presented in Figures 2, 3, Added to 3.1. Characterization of the Rhizobacteria Strains.

RESPONSE: Revised appropriately (Line 479-480)

-            Lines 447- 456: Check for Figure 6 of 4.

RESPONSE: This has been corrected and revised appropriately (Line 575-578)

 Discussion

-          Line 119: Daei et al. (2009), Hart et al. (2012) and 119 Sánchez-Matamoros et al. (2018) have.., change to style of the journal.

RESPONSE: Revised appropriately (Line 721-722)

-          Line 173: can exhibit cigent input ?????  

RESPONSE: Revised appropriately (Line 795)

 Conclusions

-       Line 178-182: delete.

RESPONSE: The sentence has been deleted as suggested by reviewer 2.

-       Lin 197: food insecurity problem. Change to the food insecurity problems

RESPONSE: Revised appropriately (Line 814)

-       Line 202: promote plant growth under drought stress and ensure food security change to promoting plant growth under drought stress and ensuring food security

RESPONSE: Revised appropriately (Line 819-820)

Comments on the Quality of English Language

The overall writing is good. However, a number of problems were detected throughout the MS, including grammar errors. please check the MS to make sure there is no language problem

RESPONSE: Authors appreciate the reviewer 2 for the brilliant review. The English Language of MS has been carefully checked and revised.

Reviewer 3 Report

Comments and Suggestions for Authors

The manuscript of Agunbiade et al. entitled “Genomic characterization of plant growth promoting-drought tolerant rhizobacteria isolated from maize plants” presents some interesting results, however it cannot be presented in the current form there are many faults. The paper, in this form, is not easy to follow. My decision is to reject this paper, but I encourage to revise the manuscript and resubmit it.

The major remarks:

1.      The title does not reflect the content of the manuscript, since the main findings are not related to genomic characterization.  Moreover, the aim of this study was not the genomic characterization of the analyzed bacteria.

2.      In my opinion, the results of this study are scarily discussed. The Authors referred to many papers but they did not discuss their own data. The aspects of novelty are difficult to find. Please indicate these aspects? Please indicate what is the novelty aspect of this study compare with the previous study of the Authors (Ojuederie and Babaola, 2023, Growth enhancement and extenuation of drought stress in maize inoculated with multifaceted ACC deaminase producing rhizobacteria)?

3.      Please explain in details how the water capacity was controlled/established/monitored? The Authors used FC (field capacity) and WC (water capacity)?  What is the difference? Please clarify. How was the water capacity assessed? Did the Authors monitor the water content/capacity before each watering?

4.      The Authors cited many papers, some of them were not included in the References list. Also, the main drawback of this manuscript is citing not the original papers just the papers in which the method was mentioned.

5.      There are some errors in the description of the experimental set-up? How was the mixed culture (A1-2+C7_8) prepared? What was the control for this experiment? How did the Authors prepare the control? How did the Authors carry out the seed inoculation? At first the Authors made the bacterial pellet and then it was wrote that the seeds were dispensed in bacterial broth culture. What were the temperature conditions and duration of this experiment? Please clarify.  This fragment is not easy to follow.

6.      There are several missing information in the descriptions of the methods. Please, see the list below.

7.      There is no data showing physicochemical parameters of the analyzed soil samples (section 2.12). Please clarify.

8.     The Authors wrote that: The combined treatment of bacteria strains (A1-2+ C7_8) displayed high plant height of 9.33 cm, 8.77 cm and 8.38 cm compared to the control low plant height of 7.27 cm, 6.08 cm and 6.10 cm at 100, 50 and 25 WC, respectively (Table 8). Similar observations were recorded for stem girth, number of leaves, leaf area and chlorophyll content. The combined treatment of strains A1-2+ C7_8 had the highest leaf numbers of 7.08, 6.50 and 6.50, stem girth of 4.58, 4.53 and 4.18 cm, chlorophyll content of 14.13, 14.64 and 13.64 at 100, 50 and 25 WC, respectively compared with the control. In terms of leaf area, highest values of 85.13, 72.37 and 73.80 cm2 at 100, 50 and 25 WC were recorded compared with the control.” (Line 438-446). It is correct according to the presented results. But there is no difference between treatment with A1-2+C7_8 versus treatment with A1-2 for stem girth, chlorophyll content, leaf area  at 50 and 25 WC. Please explain what is the advantage to apply mixed culture A1-2+C7_8 versus only A1-2?

9.     The Authors wrote: “All the treatments showed similar pattern increment in terms of plant height at 100 WC (Figure 6a). However, the  combined treatments showed an upward considerable drought tolerance response at day 25 with respect to plant length (Figures 6b and c), and stem girth (Figures 6d-f), leaf number (Figures 6g-i), chlorophyll content (Figures 6j-l) and leaf area (Figure 6m) at 100, 50 and 25 WC. Hence, the response of the maize plants to drought stress can be arranged in a decreasing order as: A1-2+ C7_8> A1-2 > C7_8 > control, corresponding to extremely effective > moderately effective > slightly effective respectively.” There is no statistical evidence that this relation is correct. Please provide the statistical proofs showing that mixed culture is better than the individual strain. The Authors should reconsider to present the data on Figure 6a-m using bar char instead of linear graph since there is no evidence that the relation number of leaf water content is linear. Moreover, the wrong Figure is cited in this fragment (Fig. 4 instead of Fig. 6). Fig.4 m does not show the weekly growth response of maize plants. Please, correct the caption of this Figure and indicate what plot (a), (b) etc. shows.  

10.  There are various marks for the same treatment; for example: combined treatment (TA+TB) while in other place it is “A1-2+C7_8” or treatment 3 etc., for the media LB and LBAGB etc. The text should be carefully corrected and the marks should be unified in the text.

11.  What was the criterion for choosing A1-2 and C7_8 for the further tests? In my opinion OD600 is not the most appropriate measurement to compare different strains? It is a great method for one strain but not the best one to compare different strains. Did the Authors measure the OD600 for the broth with various PEG concentration as a second control for each treatment?

12.  What was the idea to use different media (Luria-Bretani, Nutrient Broth, Tryptic Soy Broth and R2A broth) with different salt concentration, pH and temperature if the results obtained using these media were not compared? Please explain. It is better to have one factor (i.e. temperature) and two ones and compare the results? These media are for heterotrophic bacteria? Why did the Authors use these media?

The minor remarks:

Line 13: Polyethylene glycol: lower letter

Line 15-16: the name of species should be in normal font (A5-1, A1-2)

Line 23: bacterial inoculants

Line 116: There is something wrong with the heading. It should be corrected.

Line 118: The reference Field Ahmad et al. please follow the journal’s guidelines how to cite the paper.  Please cite the original method.

Line 118-125: There is something wrong with the description of this method. Did the Authors add Nessler reagent?

Line 121: Please correct the orthography error: “inoculated”

Line 125: Please correct the sentence: “The experiment was abbreviat in replicates”

Line 128: Provide the composition of Pikovskaya’s medium.

Line 132: Please provide the original paper in which this method was described for the first time.

Line 133: Provide the composition of the CAS medium.

Line 134: Which medium was used to prepare fresh bacterial culture?

Line 134: It should be bacterial culture instead of bacteria culture

Line 137: Please provide the original paper describing method used to assess N2 fixation. The cited paper (Ahmad, Ahmad and Khan, 2008) does not include this method.

Line 139: It should be medium instead of media

Line 140: How did the Authors prepare the bacterial culture for N2 fixation test?

Line 142: It should be Petri dishes

Line 145: Please provide the original paper according to which this test was performed. What did the Authors mean by a standardized temperature? How did the Authors prepare bacterial suspension to perform this test? There is something wrong with this description: you cannot cultivate the bacteria on solidified medium, and next obtain the supernatant. Provide the centrifuge conditions. How did you measure the intensity of IAA (=> how the calibration curve was prepared?)

Line 152: The procedure instead of procedures, “for” instead of “in”. Please correct the description of this test since something is missing.

Line 153: bacterial isolate

Line 160: provide the composition of minimal medium.

Line 159-166: Please rewrite this test since it is difficult to repeat this test following this description. 

Line 161: is should be rather supplemented

Line 169: provide the manufacturer name of Zymo DNA extraction kit

Section 2.8.2: Provide the references for the programs applied in this analysis (Bio-Edit, MEGA-X etc). Also provide the versions of these programs. How many sequences were used to build these tree? What about the consensus tree?

Section 2.9.1. Please correct the description of this test. There is something wrong. Which medium was used? Explain LB abbreviation. Which reagents were used to obtain ph 4 and 10? Please rewrite this fragment: “Each bacteria overnight culture was divided exactly 20ml find out please l” , and “The bacterial isolate was 212 inoculated in triplicates, and the inoculated LB broth (…)”

Section 2.9.2. The remarks as for section 2.9.1. Please rewrite this description. Which medium was used in this test? Please correct: “25 ml of LB broth were gently vortexed after 5 l correctof each bacterial strains was added.” What did the Authors mean in this sentence: “The inoculated broth from each media was incubated at (25, 30, 35, and 40°C) and performed in triplicates.”? Why did the Author measure OD 600 after 24-hour incubation in the temperature test (2.9.2) while in the pH test (2.9.1) the measurement was performed after 48-hour incubation?

Section 2.9.3 Why did the Authors applied 3 different media? All of these media cover mostly the heterotrophic bacteria? Please explain, what was the aim to apply very similar media to test the tolerance towards different salt concentrations, heavy metal, pH and temperature?

Section 2.9.4: Why did the Authors apply only one concentration of heavy metal salt (100 mg/l)?

Line 251: explain the CRD abbreviation

Line 262: Please rewrite: “A total of 192 pots containing two maize plants were produced by setting 262 up twenty 24 treatments with 4 replicates per treatment.” 20 or 24?

Line 288: dry shoot mass it should be abbreviated as D.S.M (there is already D.R.M)

Section 2.11.1. Explain the RWC of leaves measurement and correct the formula. Explain the abbreviations.

Section 2.11.3 Please rewrite this section, since there are some text duplications.

Line 317, 322: Please correct it was maize leaves

Line 320 it should be described by…

Line 321: Please remove one “by”

Line 338: Please remove one “and clay”

Line 342: Please remove “DELETE”

Line 358-361 Please move this sentence to the Methods Section (for example to subsection: 2.8.2. “Alignement Sequence and …” or consider to create additional section like Data availability

Figure 1. Please provide the strain and the NCB accessing number of the closed match for the analyzed strains (blastn analysis).  Please provide the scale bar for each picture since it seems that it was recorded using different magnification. Did the Authors use two different microscopes to record the photos of gram-stained bacteria? Why is the yellow background on some photos (it does not look like the photo from optical microscope)? Aeromonas veroni C7_8 is not gram + bacteria, it is gram – bacteria. So, it should pink on microscopy using the Gram stain.  Is Bacillus halotolerans A9-10 contaminated by another bacteria (it looks like there are two different cell type)? Bacillus halotolerans A9-10 should have the bacilli shape, the not cocci shape. Why does A9-10 have the cocci shape?

Line 392: Please add the strain identifier/number to the Table 3.

Table 3: Did the Authors assess nitrogen production? In the Methods the Authors wrote about nitrogen fixation not production. Please clarify. Nitrogen production is not equal nitrogen fixation.

Line 380-382: Please rewrite the sentences: (…) exhibited N2 fixation potential, produced siderophore, ammonia, IAA, ACC and solubilize phosphate. 

Remarks for Figure 2 and Figure 3:

1)      Please provide the scale bar for these trees.

2)      Please provide the NCBI accession number for the strains presented in the Figure.

3)      Where is the outgroup?

Table 4: What does LBAGB abbreviation mean? In the text it is LBB as Luria-Bretani Broth

Line 412: The data in Table 5 do not present the pH of amended media. Please, rewrite this sentences.

Line 414: Please rewrite this sentence since it does not reflect the difference in the pH avlues..

Table 7: The name of PB2+ should be changed, it is Pb2+. There is some mistakes: in the methods section it was written that CdSO4 ·8H2O and K2Cr2Owere used to evaluate metal tolerance of bacteria; so it was Cd2+ and Cr2O72- (Cr(VI)) used in this study not Cdand Cr2+. On the other hand, there is no compounds like Cr(II) and Cd(I).

Line 438: FC or WC?

Figure 4 m. In my opinion this Figure does not present the weekly growth response of maize pants. Please, reconsider this Figure.

Line 26 (section 3.7): co-inoculation

Line 70-71(section Discussion): The references are missing.

Please, check the papers cited in the text with the References list:

Reference no 37: correct the surname of the Author.

Comments on the Quality of English Language

Please correct the English language of the text since there are some word duplications and ortographical errors.

Author Response

AUTHORS’ RESPONSE TO REVIEWER 3

Comments and Suggestions for Authors

The manuscript of Agunbiade et al. entitled “Genomic characterization of plant growth promoting-drought tolerant rhizobacteria isolated from maize plants” presents some interesting results, however it cannot be presented in the current form there are many faults. The paper, in this form, is not easy to follow. My decision is to reject this paper, but I encourage to revise the manuscript and resubmit it.

RESPONSE: Authors’ appreciate reviewer 3 for the brilliant review and contributively suggestions. We are grateful.

The major remarks:

  1. The title does not reflect the content of the manuscript, since the main findings are not related to genomic characterization.  Moreover, the aim of this study was not the genomic characterization of the analyzed bacteria.

RESPONSE: The title of the manuscript has been modified and revised (Line 2-3).

  1. In my opinion, the results of this study are scarily discussed. The Authors referred to many papers but they did not discuss their own data. The aspects of novelty are difficult to find. Please indicate these aspects? Please indicate what is the novelty aspect of this study compare with the previous study of the Authors (Ojuederie and Babaola, 2023, Growth enhancement and extenuation of drought stress in maize inoculated with multifaceted ACC deaminase producing rhizobacteria)?

RESPONSE: The manuscript has been critically checked and revised. The novelty of this study has been revised and stated (Line 88-95).

  1. Please explain in details how the water capacity was controlled/established/monitored? The Authors used FC (field capacity) and WC (water capacity)?  What is the difference? Please clarify. How was the water capacity assessed? Did the Authors monitor the water content/capacity before each watering?

RESPONSE: In fact, in a controlled environment, the maize plants were grown in culture pots. To water these plants, we need to know exactly how much water the soil can absorb, so we determined the maximum water-holding capacity of the studied soil. After determining this, we used 2/9 of this value to water the plants at every 48 to 72 hours.

Did the authors use FC (field capacity) and WC (water capacity)? 

What is the difference? Please specify.

We apologize for the error. This part has been corrected in the new document.

How was water capacity assessed?

Rather, the maximum water-holding capacity of the soil was determined before watering the plants in the pots, so as not to exceed the amount of water required.

  1. The Authors cited many papers, some of them were not included in the References list. Also, the main drawback of this manuscript is citing not the original papers just the papers in which the method was mentioned.

RESPONSE: All the cited references were included and revised accordingly. The original papers cited in the method section have been thoroughly revised.

  1. There are some errors in the description of the experimental set-up? How was the mixed culture (A1-2+C7_8) prepared? What was the control for this experiment? How did the Authors prepare the control? How did the Authors carry out the seed inoculation? At first the Authors made the bacterial pellet and then it was wrote that the seeds were dispensed in bacterial broth culture. What were the temperature conditions and duration of this experiment? Please clarify.  This fragment is not easy to follow.

RESPONSE: The comments have been addressed and revised appropriately (Line 348, 352-357, 360).

  1. There are several missing information in the descriptions of the methods. Please, see the list below.

RESPONSE: The missing information has been carefully checked and revised in the methods section.

  1. There is no data showing physicochemical parameters of the analyzed soil samples (section 2.12). Please clarify.

RESPONSE: This section has been expunged from the manuscript.

  1. The Authors wrote that: “The combined treatment of bacteria strains (A1-2+ C7_8) displayed high plant height of 9.33 cm, 8.77 cm and 8.38 cm compared to the control low plant height of 7.27 cm, 6.08 cm and 6.10 cm at 100, 50 and 25 WC, respectively (Table 8). Similar observations were recorded for stem girth, number of leaves, leaf area and chlorophyll content. The combined treatment of strains A1-2+ C7_8 had the highest leaf numbers of 7.08, 6.50 and 6.50, stem girth of 4.58, 4.53 and 4.18 cm, chlorophyll content of 14.13, 14.64 and 13.64 at 100, 50 and 25 WC, respectively compared with the control. In terms of leaf area, highest values of 85.13, 72.37 and 73.80 cm2at 100, 50 and 25 WC were recorded compared with the control.” (Line 438-446). It is correct according to the presented results. But there is no difference between treatment with A1-2+C7_8 versus treatment with A1-2 for stem girth, chlorophyll content, leaf area  at 50 and 25 WC. Please explain what is the advantage to apply mixed culture A1-2+C7_8 versus only A1-2?

RESPONSE: The activity of the bacteria can be linked to the prevailing conditions of the growth medium, which might results in no significant or slight significant differences in the results obtained.

  1. The Authors wrote: “All the treatments showed similar pattern increment in terms of plant height at 100 WC (Figure 6a). However, the  combined treatments showed an upward considerable drought tolerance response at day 25 with respect to plant length (Figures 6b and c), and stem girth (Figures 6d-f), leaf number (Figures 6g-i), chlorophyll content (Figures 6j-l) and leaf area (Figure 6m) at 100, 50 and 25 WC. Hence, the response of the maize plants to drought stress can be arranged in a decreasing order as: A1-2+ C7_8> A1-2 > C7_8 > control, corresponding to extremely effective > moderately effective > slightly effective respectively.” There is no statistical evidence that this relation is correct. Please provide the statistical proofs showing that mixed culture is better than the individual strain. The Authors should reconsider to present the data on Figure 6a-m using bar char instead of linear graph since there is no evidence that the relation number of leaf water content is linear. Moreover, the wrong Figure is cited in this fragment (Fig. 4 instead of Fig. 6). Fig.4 m does not show the weekly growth response of maize plants. Please, correct the caption of this Figure and indicate what plot (a), (b) etc. shows.  

RESPONSE: Now Figure 3. It has been revised appropriately (Line 596).

  1. There are various marks for the same treatment; for example: combined treatment (TA+TB) while in other place it is “A1-2+C7_8” or treatment 3 etc., for the media LB and LBAGB etc. The text should be carefully corrected and the marks should be unified in the text.

RESPONSE: This has been corrected and revised (Line 535-551; 639-656).

  1. What was the criterion for choosing A1-2 and C7_8 for the further tests? In my opinion OD600is not the most appropriate measurement to compare different strains? It is a great method for one strain but not the best one to compare different strains. Did the Authors measure the OD600 for the broth with various PEG concentration as a second control for each treatment?

RESPONSE: The criterion for selecting the rhizobacterial strains has been corrected revised (Line 483-486).

  1. What was the idea to use different media (Luria-Bretani, Nutrient Broth, Tryptic Soy Broth and R2A broth) with different salt concentration, pH and temperature if the results obtained using these media were not compared? Please explain. It is better to have one factor (i.e. temperature) and two ones and compare the results? These media are for heterotrophic bacteria? Why did the Authors use these media?

RESPONSE: The media were used based on the research design by the authors and the idea was to determine which media could best be suitable for further studies (Line 236, 249, 259, 270).

The minor remarks:

Line 13: Polyethylene glycol: lower letter

RESPONSE: Revised appropriately (Line 13).

Line 15-16: the name of species should be in normal font (A5-1, A1-2)

RESPONSE: Revised appropriately (Line 15-17).

Line 23: bacterial inoculants

RESPONSE: Revised appropriately (Line 23).

Line 116: There is something wrong with the heading. It should be corrected.

RESPONSE: Corrected and revised appropriately (Line 145).

Line 118: The reference Field Ahmad et al. please follow the journal’s guidelines how to cite the paper.  Please cite the original method.

RESPONSE: The reference has been expunged and revised appropriately (Line 149).

Line 118-125: There is something wrong with the description of this method. Did the Authors add Nessler reagent?

RESPONSE: The method has been corrected and revised appropriately (Line 152-154).

Line 121: Please correct the orthography error: “inoculated”

RESPONSE: Revised appropriately (Line 151).

Line 125: Please correct the sentence: “The experiment was abbreviat in replicates”

RESPONSE: Revised appropriately (Line 156-157).

Line 128: Provide the composition of Pikovskaya’s medium.

RESPONSE: The composition of the Pikovskaya’s medium has been provided and revised appropriately (Line 162-166).

Line 132: Please provide the original paper in which this method was described for the first time.

RESPONSE: The original paper for this method has been provided and revised appropriately (Line 173).

Line 133: Provide the composition of the CAS medium.

RESPONSE: The composition of the CAS medium has been provided and revised appropriately (Line 174-176).

Line 134: Which medium was used to prepare fresh bacterial culture?

RESPONSE: The medium used was nutrient agar (Line 178).

Line 134: It should be bacterial culture instead of bacteria culture

RESPONSE: Revised appropriately (Line 177).

Line 137: Please provide the original paper describing method used to assess N2 fixation. The cited paper (Ahmad, Ahmad and Khan, 2008) does not include this method.

RESPONSE: The original paper for this method has been provided and revised appropriately (Line 185).

Line 139: It should be medium instead of media

RESPONSE: Revised appropriately (Line 185).

Line 140: How did the Authors prepare the bacterial culture for N2 fixation test?

RESPONSE: Revised appropriately (Line 187-189).

Line 142: It should be Petri dishes

RESPONSE: Revised appropriately (Line 190).

Line 145: Please provide the original paper according to which this test was performed. What did the Authors mean by a standardized temperature? How did the Authors prepare bacterial suspension to perform this test? There is something wrong with this description: you cannot cultivate the bacteria on solidified medium, and next obtain the supernatant. Provide the centrifuge conditions. How did you measure the intensity of IAA (=> how the calibration curve was prepared?)

RESPONSE: The section has been carefully checked and revised with the original paper provided (Line 194-201).

Line 152: The procedure instead of procedures, “for” instead of “in”. Please correct the description of this test since something is missing.

RESPONSE: ‘for’ has been revised (Line 205). The description of this method has been revised appropriately (Line 208-212).

Line 153: bacterial isolate

RESPONSE: Revised appropriately (Line 207).

Line 160: provide the composition of minimal medium.

RESPONSE: The minimal medium composition has been provided and revised appropriately (Line 223-228).

Line 159-166: Please rewrite this test since it is difficult to repeat this test following this description. 

RESPONSE: The entire description for the test has been corrected and revised (Line 216-232).

Line 161: is should be rather supplemented

RESPONSE: This has been revised accordingly.

Line 169: provide the manufacturer name of Zymo DNA extraction kit

RESPONSE: The manufacturer name of Zymo DNA extraction kit has been provided and revised (Line 283-284).

Section 2.8.2: Provide the references for the programs applied in this analysis (Bio-Edit, MEGA-X etc). Also provide the versions of these programs. How many sequences were used to build these tree? What about the consensus tree?

RESPONSE: This has been corrected and revised (Line 319-320).

Section 2.9.1. Please correct the description of this test. There is something wrong. Which medium was used? Explain LB abbreviation. Which reagents were used to obtain ph 4 and 10? Please rewrite this fragment: “Each bacteria overnight culture was divided exactly 20ml find out please l” , and “The bacterial isolate was 212 inoculated in triplicates, and the inoculated LB broth (…)”

RESPONSE: The entire description for the test has been corrected and revised (Line 238-241).

Section 2.9.2. The remarks as for section 2.9.1. Please rewrite this description. Which medium was used in this test? Please correct: “25 ml of LB broth were gently vortexed after 5 l correctof each bacterial strains was added.” What did the Authors mean in this sentence: “The inoculated broth from each media was incubated at (25, 30, 35, and 40°C) and performed in triplicates.”? Why did the Author measure OD 600 after 24-hour incubation in the temperature test (2.9.2) while in the pH test (2.9.1) the measurement was performed after 48-hour incubation?

RESPONSE: The entire description for the test has been corrected and revised (Line 249-254). The incubation time of 24 hours has been revised (Line 243, 254).

Section 2.9.3 Why did the Authors applied 3 different media? All of these media cover mostly the heterotrophic bacteria? Please explain, what was the aim to apply very similar media to test the tolerance towards different salt concentrations, heavy metal, pH and temperature?

RESPONSE: The media were used based on the research design by the authors and the idea was to determine which media could best be suitable for further studies (Line 236, 249, 259, 270).

Section 2.9.4: Why did the Authors apply only one concentration of heavy metal salt (100 mg/l)?

RESPONSE: The only one concentration of heavy metal salt (100 mg/l) was used based on the reference to the literature as a guide (Line 272)

Line 251: explain the CRD abbreviation

RESPONSE: The sentence has been expunged and revised (Line 323-324).

Line 262: Please rewrite: “A total of 192 pots containing two maize plants were produced by setting 262 up twenty 24 treatments with 4 replicates per treatment.” 20 or 24?

RESPONSE: ’24 treatments’ has been revised (Line 343).

Line 288: dry shoot mass it should be abbreviated as D.S.M (there is already D.R.M)

RESPONSE: Revised appropriately (Line 373).

Section 2.11.1. Explain the RWC of leaves measurement and correct the formula. Explain the abbreviations.

RESPONSE: Revised appropriately (Lines 387-393).

Section 2.11.3 Please rewrite this section, since there are some text duplications.

RESPONSE: Now 2.9.3. This section has been re-written and revised appropriately (Line 411-424).

Line 317, 322: Please correct it was maize leaves

RESPONSE: Revised appropriately (Line 411).

Line 320 it should be described by…

RESPONSE: Revised appropriately (Line 417).

Line 321: Please remove one “by”

RESPONSE: Revised appropriately (Line 417).

Line 338: Please remove one “and clay”

RESPONSE: The sentence has been expunged from the manuscript.

Line 342: Please remove “DELETE”

RESPONSE: The sentence has been expunged from the manuscript.

Line 358-361 Please move this sentence to the Methods Section (for example to subsection: 2.8.2. “Alignement Sequence and …” or consider to create additional section like Data availability

RESPONSE: The data availability section has been added and revised as suggested (457-461).

Figure 1. Please provide the strain and the NCB accessing number of the closed match for the analyzed strains (blastn analysis).  Please provide the scale bar for each picture since it seems that it was recorded using different magnification. Did the Authors use two different microscopes to record the photos of gram-stained bacteria? Why is the yellow background on some photos (it does not look like the photo from optical microscope)? Aeromonas veroni C7_8 is not gram + bacteria, it is gram – bacteria. So, it should pink on microscopy using the Gram stain.  Is Bacillus halotolerans A9-10 contaminated by another bacteria (it looks like there are two different cell type)? Bacillus halotolerans A9-10 should have the bacilli shape, the not cocci shape. Why does A9-10 have the cocci shape?

RESPONSE: All the comments have been revised appropriately as Supplementary file attachment.

Line 392: Please add the strain identifier/number to the Table 3.

RESPONSE: Now Table 2. Revised appropriately (Line 505).

Table 3: Did the Authors assess nitrogen production? In the Methods the Authors wrote about nitrogen fixation not production. Please clarify. Nitrogen production is not equal nitrogen fixation.

RESPONSE: Revised appropriately (Line 505).

Line 380-382: Please rewrite the sentences: (…) exhibited N2 fixation potential, produced siderophore, ammonia, IAA, ACC and solubilize phosphate. 

RESPONSE: Revised appropriately (Line 495-497).

Remarks for Figure 2 and Figure 3:

1)      Please provide the scale bar for these trees.

2)      Please provide the NCBI accession number for the strains presented in the Figure.

3)      Where is the outgroup?

RESPONSE: Now Figure 1 and 2. The Figures have been revised and modified (Line 509, 514).

Table 4: What does LBAGB abbreviation mean? In the text it is LBB as Luria-Bretani Broth

RESPONSE: The abbreviations have been corrected and revised (Line 535-551).

Line 412: The data in Table 5 do not present the pH of amended media. Please, rewrite this sentences.

RESPONSE: Now Table 4. The sentence has been rewritten and revised appropriately (Line 525-527).

Line 414: Please rewrite this sentence since it does not reflect the difference in the pH avlues.

RESPONSE: The sentence has been rewritten and revised appropriately (Line 525-527).

Table 7: The name of PB2+ should be changed, it is Pb2+. There is some mistakes: in the methods section it was written that CdSO4 ·8H2O and K2Cr2Owere used to evaluate metal tolerance of bacteria; so it was Cd2+ and Cr2O72- (Cr(VI)) used in this study not Cdand Cr2+. On the other hand, there is no compounds like Cr(II) and Cd(I).

RESPONSE: This has been corrected and revised (Line 272, 533-534).

Line 438: FC or WC?

RESPONSE: ‘FC’ revised (Line 563).

Figure 4 m. In my opinion this Figure does not present the weekly growth response of maize pants. Please, reconsider this Figure.

RESPONSE: Now Figure 3. This has been reconstructed and revised appropriately (Line 596).

Line 26 (section 3.7): co-inoculation

RESPONSE: Revised appropriately (Line 619).

Line 70-71(section Discussion): The references are missing.

RESPONSE: References have been added and revised (Line 672).

Please, check the papers cited in the text with the References list:

Reference no 37: correct the surname of the Author.

RESPONSE: This has been revised appropriately.

Comments on the Quality of English Language

Please correct the English language of the text since there are some word duplications and ortographical errors.

RESPONSE: The English Language check of the manuscript has been carefully checked and revised.

Reviewer 4 Report

Comments and Suggestions for Authors

The manuscript identified and characterized Rhizobacteria strains, demonstrated plant growth-promoting capabilities and drought stress response tolerance the Rhizobacteria strains. The phenotypical studies are comprehensive and scientifically sound.

I have minor revisions for the authors to improve the quality of the manuscript.

  1. In intro section, spell out ACC deaminase
  1. Section 3.4 and 3.5 have data-heavy tables. I would like to suggest switch to heat maps or bar graphs to highlight the difference.

Comments on the Quality of English Language

I found a few sentences hard to read. Please consider plain language. For example, line 34: The narrow soil region, which directly influenced by root secretions is termed the rhizosphere, while the associated soil microbes jointly make up the root microbiome. This can be broken into 2 sentences.

Author Response

AUTHORS’ RESPONSE TO REVIEWER 4

Comments and Suggestions for Authors

The manuscript identified and characterized Rhizobacteria strains, demonstrated plant growth-promoting capabilities and drought stress response tolerance the Rhizobacteria strains. The phenotypical studies are comprehensive and scientifically sound.

RESPONSE: Authors’ appreciate reviewer 4 for finding their manuscript comprehensive and scientifically sound.

I have minor revisions for the authors to improve the quality of the manuscript.

  1. In intro section, spell out ACC deaminase

RESPONSE: The ACC deaminase has been spelt out and revised appropriately (Line 18, 62-63).

  1. Section 3.4 and 3.5 have data-heavy tables. I would like to suggest switch to heat maps or bar graphs to highlight the difference.

RESPONSE: Figure 3 has been presented in bar graphs (Line 596).

Comments on the Quality of English Language

I found a few sentences hard to read. Please consider plain language. For example, line 34: The narrow soil region, which directly influenced by root secretions is termed the rhizosphere, while the associated soil microbes jointly make up the root microbiome. This can be broken into 2 sentences.

RESPONSE: The sentence has been revised appropriately (Line 48-49).

Round 2

Reviewer 2 Report

Comments and Suggestions for Authors

Accept in the present form

Author Response

A cover letter explaining, point by point, the details
of the revisions to the manuscript and author’s responses to the referees’
comments.

AUTHORS’ RESPONSE TO REVIEWER 2

Comments and Suggestions for Authors

Article title:

“Genomic Characterization of Plant Growth Promoting-Drought Tolerant Rhizobacteria Isolated from Maize Plants".

The work is unique, really interesting, and valid for publication after some modification points:   Major comments:

Authors’ RESPONSE: Authors’ appreciate reviewer 2 for the brilliant review and finding our manuscript to be unique, interesting and valid for publication.

Reviewer’s Comment:  During the isolation, three media (Luria agar, Nutrient agar, and Tryptic agar), were used. Then, one media (NA) was used in evaluations of growth regulators and drought tolerance. Why?

Authors’ RESPONSE:  The NA was used because of high microbial counts recorded compared to other media. Hence, it was used for other tests (Line 116-117).

Reviewer’s Comment: Added 2.9. Tolerance of bacterial isolates to pH, temperatures, salt concentrations, and heavy metals before 2.8. Extraction of DNA and Genotypic Identification. Rearrange again in all the sections of the manuscript.

Authors’ RESPONSE: This has been rearranged and revised appropriately (Line 234-320).

 Reviewer’s Comment: What about "The growth media used" in Bacterial tolerance to different pH (Line 207), different temperatures (Line 216), different salt concentrations (Line 224), and different heavy metals (Line 234)?

Authors’ RESPONSE: The growth media used has been revised appropriately (Line 236-237; 249-250; 258-259, 270-271).

The Nutrient broth, Luria-Bertani Broth, Tryptic Soy Broth, and Reasoner’s 2A Broth were used for this test. (Line 236-237)

The Nutrient broth, Luria-Bertani Broth, Tryptic Soy Broth, and Reasoner’s 2A Broth were used for the test. The broth medium was prepared and sterilized at 121°C for 15 minutes. 25 ml Luria Bertani broth was gently vortexed after each bacterial strain was inoculated. The inoculated broth from each media was incubated at 25, 30, 35, and 40°C. A spectrophotometer (Thermo Spectronic; Meck, South Africa) was used to measure the bacterial growth’s optical density at 600 nm after 24 hours of incubation. Line (249-254).

The Nutrient broth, Luria-Bertani Broth, Tryptic Soy Broth, and Reasoner’s 2A Broth were used for the test; (Line 258-259,)

Reviewer’s Comment: Why was the anion radical not proven in the experiment? Line 236.

Authors’ RESPONSE: The Nutrient broth, Luria-Bertani Broth, Tryptic Soy Broth, and Reasoner’s 2A Broth were prepared and sterilized at 121°C for 15 minutes. The media were supplemented with 100 mg/l of various heavy metals, Pb2+, Cr2O72- and Cd2+ in test tubes to test for the tolerance of each bacterial isolate to various heavy metals. This has been revised appropriately in the manuscript (Line 272).

Reviewer’s Comment:  Line 249: and sterilised for 15 minutes at 121°C. How it’s a soil (it's better to sterilize to 1 h).

Authors’ RESPONSE:  The sieved soil samples were sealed in autoclavable plastic bags and sterilized for 1 hour at 121°C (autoclave temperature). To ensure complete sterilization, this process was repeated three times. This has been revised appropriately (Line 330).

Reviewer’s Comment: How to calculate the FC (100, 50, 25%)?

Authors’ RESPONSE: The calculation of Field Capacity (FC)

Add 6 kg of soil 3 Litters of Maximum Water Rentention Capapcity (MRC), i.e. 50 ml MRC water for 100 g of soil.

2/9 MRC water was used. i.e. (3000 ml x 2)/9 = 666, 66 ml or 670 ml.

For drought:

100%, 670 ml of water for 6 kg of soil was used.

50%, 335 ml of water for 6 kg of soil was used.

25%, 167 ml, 5, or 170 ml of water for 6 kg of soil was used.

This has been added to the Supplementary file document (Line 333).

Reviewer’s Comment: Added 2.12. Physicochemical Parameter Analyses of Soil Samples in the first MM section.

Authors’ RESPONSE: This section has been expunged from the body of the manuscript.

Reviewer’s Comment: Added 3.1. Characterization of the Rhizobacteria Strains in supplementary file (Figure 1, Table 1).

Authors’ RESPONSE: This has been revised and taken to supplementary file (465).

Reviewer’s Comment:  It would have been better to perform tolerance to drought stress under different growth conditions tests for all tested isolates, as is the case with growth regulators (PGP)

Authors’ RESPONSE: Authors appreciate the reviewer 2 suggestion, which is remarkable and contributively. However, the scope of this research was as presented. Nevertheless, the suggestions shall be considered in our future studies.

    Minor comments: Abstract:

Reviewer’s Comment:  Line 10: Bioinucula change to bioinocula

Authors’ RESPONSE: bioinocula. Revised appropriately in the manuscript (Line 10)

Reviewer’s Comment: Line 15 and 16: Change Bacillus to B.

Authors’ RESPONSE: Revised appropriately in the manuscript(Lines 15-17)

Reviewer’s Comment:  Line 21: combined. An increase change to combined. An increased

Authors’ RESPONSE: Aeromonas strains, A1-2 and C7_8 showing the highest drought tolerance of 0.71 and 0.77, respectively were selected for the bioinoculation, singularly and combined. An increase in the above and below-ground biomass of the maize plants at 100, 50, and 25% water holding capacity (WHC) was recorded. Revised appropriately (Lines 22)

 Introduction: Acceptable

Authors’ RESPONSE: Authors appreciate Reviewer 2 for accepting our Introduction section as presented. However, authors have revised this section based on other reviewers’ comments.

 Materials and methods:

Reviewer’s Comment:  Line 110: add CAS No of Lot No of the chemical used of PEG8000.

Authors’ RESPONSE: Evaluation of the bacterial isolates’ drought tolerance was carried out in a nutrient-broth medium amended with varied PEG8000 concentrations (0, 5, 10, 15, 20, 25, and 30%). The PEG800 has CAS number (25322683) and Catalog number (P0131). Revised appropriately in the manuscript  (Line 138)

Reviewer’s Comment:  Line 111: (0%, 5%, 10%, 15%, 20%, 25%, 30%). Change to (0, 5, 10, 15, 20, 25 and 30%).

Authors’ RESPONSE: (0, 5, 10, 15, 20, 25, and 30%). Revised appropriately in the manuscript (Line 137)

Reviewer’s Comment:   Line 112: delete distance " continuous    shaking ".

Authors’ RESPONSE:  under continuous shaking .Revised appropriately (Line 140)

Reviewer’s Comment:  Line 121-122: A rocking abbreviat was used for incubation at 28±2°C for 4 days. was poured into each of the tubes and allow to stay for 5 minutes. (Rewrite again).

Authors’ RESPONSE: incubated at 28±2°C for 4 days on a rotary shaker (SI-600, LAB Companion, Korea) at 120 rpm. 0.5 ml of Nessler’s reagent was added to each test tube and allowed to stay for 5 minutes for color development.  The statement has been revised appropriately in the manuscript (Line 152-154)

Reviewer’s Comment:  Line 140: MgSO47H2O 0.5 g; NaCI 0.5 g; change to MgSO4 7H2O 0.5 g; NaCl 0.5 g;

Authors’ RESPONSE: MgSO4 7H2O 0.5 g; NaCl 0.5 g; Changed and revised appropriately (Line 185-186)

Reviewer’s Comment:  Line 140: Complete the component of the media used such as pH, and sterilized conditions. Etc

Authors’ RESPONSE: The rhizobacteria nitrogen fixation characteristics were determined according to AlAli, et al. [33] in Jensen’s medium composed of sucrose 20 g; K2HPO4 1.0 g; MgSO4.7H2O 0.5 g; NaCl 0.5 g; FeSO4.7H2O 0.1 g; CaCO3 2.0 g; Agar 15.0 g; Na2MoO4 0.005 g; and 1 Litre sterile distilled water. The medium was adjusted to pH 7.2 before sterilization at 121oC for 15 minutes. Twenty-four-hour-old rhizobacterial cultures were inoculated into the Petri plates containing Jensen’s medium and incubated at 28±2oC for 7 days. Revised appropriately in the manuscript (Line 185-189)

Reviewer’s Comment:   Line 146: tryptophan (L or D)

Authors’ RESPONSE: L-tryptophan. Revised appropriately  in the manuscript (Line 195)

Reviewer’s Comment:  Line 211: please l, ?????

Authors’ RESPONSE: 20 ml of each bacteria overnight culture was divided equally and then added to sterile 10 ml LB broth and homogenized.  Revised appropriately in the manuscript (Line 240)

Reviewer’s Comment:  Line 244: Added the design of the experiment in Experimental Set-up and Maize Growth Conditions

Authors’ RESPONSE: The experimental set-up for maize growth under a greenhouse was achieved using a three-factorial complete randomized design (CRD). Revised appropriately in the manuscript (Line 323-324)

Reviewer’s Comment:   Line 251: A. caviae, write completely.

Authors’ RESPONSE: To ensure complete sterilization, this process was repeated three times. This sentence has been expunged and revised appropriately from Line 331)

Reviewer’s Comment:   Line 262: twenty 24 treatments???

Authors’ RESPONSE: A total of 192 pots containing two maize plants were produced by setting up 24 treatments with 4 replicates per treatment. Revised appropriately in the manuscript (Line 342-343)

Reviewer’s Comment:  Line 262 and 263: Rewrite again

Authors’ RESPONSE: A total of 192 pots containing two maize plants were produced by setting up 24 treatments with 4 replicates per treatment. The sentence has been rewritten and revised appropriately (Line 342-343)

Reviewer’s Comment:  Line 301: RWC measured, what is the time?

Authors’ RESPONSE: The leaf-relative water content was determined according to the methods described by Ortiz, et al. [42] with little modifications, and the ‘youngest fully developed leaves of each plant’ were used. Fresh leaf samples were weighed (Fresh weight-FW) and placed in test tubes saturated with water and kept at 4°C for 48 hours. Thereafter, the leaf samples were weighed again to obtain the turgid weight (TW) and oven-dried at 60°C for 24 hours, and dry weights (DW) were obtained. The leaf-relative water content was calculated as follows: The sentence has been revised appropriately in the manuscript (Line 387-393)

Reviewer’s Comment: Line 303: added the number for the formula.

Authors’ RESPONSE: Leaf relative water content (%)  =             (FW−DW).         x 100

                                                                                                         (TW−DW).

  The information of the formula has been well presented and revised appropriately in the manuscript (Line 395)

Reviewer’s Comment:   Line 306: From each of the eight replicates for each treatment????   8 or 4 replicates.

Authors’ RESPONSE:  Four replicates for each treatment. This have been revised appropriately in the manuscript (Line 401)

Reviewer’s Comment:  Line 315: added the number for the formula.

Authors’ RESPONSE:  The following formula is for  electrolyte leakage:  Electrolyte leakage (%)  x 100%. Revised appropriately in the manuscript (Line 409)

Reviewer’s Comment: Line 317: Soluble sugars in soybean leaves were extracted. Oh, Soybean or maize?????

Authors’ RESPONSE: Soluble Sugar Content in Maize Leaves were extracted. Revised appropriately in the manuscript (Line 411)

Reviewer’s Comment:   Line 317 -328: Rewrite again

Authors’ RESPONSE: 2.9.3 Soluble Sugar Content in Maize Leaves

One hundred mg (100 mg) of maize leaves were homogenized in 4 ml of 80% ethanol and centrifuged for 10 minutes at 5000 rpm. 1 ml of the supernatant was mixed with 5 ml of concentrated sulphuric acid (H2SO4) and 1 ml of 5% phenol. Samples were vigorously agitated and allowed to cool for 30 minutes and absorbance was measured at 485 nm using a spectrophotometer (ThermoSpectronic, South Africa). The concentration of soluble sugar was calculated from a standard curve ‘established with a reference glucose solution’ as described by Zarik, et al. [43] with little modifications. Briefly, stock solutions of glucose were prepared by weighing 10, 20, 30, 40, 50, 60, and 70 mg of D(+)-glucose in 10 ml of sterile distilled water. 2 ml of each stock solution was pipetted into 3 cleaned test-tubes. 1 ml of 5% phenol was added. 5 ml of concentrated H2SO4 was quickly added. The tubes were allowed to stand for 10 minutes, then shaken and placed in a water bath for 10 min at 51-57°C. Tubes were allowed to cool for 30 minutes, and absorbance was measured at 485 nm using a spectrophotometer and a standard curve was plotted from the values obtained. This sub-section has been re-written and revised appropriately in the manuscript (Line 410-424)

Reviewer’s Comment:  Line 330: Proline in maize leaves measured, what the time?

Authors’ RESPONSE: 2.9.4 Proline Content in Maize Leaves

Proline in maize leaves was extracted using Bates, et al. [44] technique with only minor modifications. Briefly, 1.25 g of ninhydrin was dissolved in 20 ml of 6 M phosphoric acid and 30 ml of glacial acetic acid by heating on a hot-plate with agitation. The solution was allowed to cool and kept at 4°C and the solution became stable after 24 hours. Approximately 1 g of fresh maize leaf sample was ground in 10 ml of 3% aqueous sulfo-salicyclic acid and centrifuged at 10000 g for 10 minutes. 2 ml of the supernatant was reacted with 2 ml of glacial acetic acid and 2 ml of acid-ninhydrin solution in 45 ml falcon tubes at 100°C in a water bath for 60 minutes and the reaction was stopped in an ice box. 4 ml toluene was added to extract the mixture and agitated vigorously for 15-20 seconds in a shaker incubator at 250 rpm. The mixture was kept in the dark for 30 minutes, the ‘chromophore containing toluene was aspirated from the aqueous phase and the absorbance was read at 520 nm using toluene for a blank. The concentration of proline was estimated from a standard curve ‘established with a reference proline solution. Briefly, a 1 mg/ml stock solution of proline was prepared by weighing 10 mg of proline (DL-Proline, China) in 10 ml of sterile water. 0, 50, 100, 150, 200, 250, and 300 μl of the stock solution was pipetted into seven tubes containing 300, 250, 200, 150, 100, 50, and 0 μl of sterile water, respectively. The mixtures were then reacted with 2 ml of glacial acetic acid and 2 ml of acid-ninhydrin solution in 45 ml falcon tubes at 100°C in a water bath for 60 minutes and the reaction was stopped in an ice box. The mixture was vigorously agitated using a vortex (Vortex Genie, U.S.A) after adding 4 ml of toluene. The mixture was kept in the dark for 30 minutes and the absorbance of the proline-containing upper layer was read at 520 nm using toluene for a blank’ and the proline standard curve was plotted from the absorbance values. This sub-section has been revised appropriately in the manuscript (Line 426-449)

Reviewer’s Comment: Line 342: DELETE ?????.

Authors’ RESPONSE: This has been expunged from the manuscript.

 Results

Reviewer’s Comment: Table 2: LSD for 1%, 5% or….? Added

Authors’ RESPONSE: Table 1. Response of Rhizobacterial Strains to Drought Stress Amended with Various Concentrations of PEG-8000.

Treatment

Control

5%

10%

15%

20%

25%

30%

A5-1

0.35 ± 0.17fg

0.18 ±0.01c

0.16±0.01d

0.16 0.00f

0.16± 0.00d

0.13±0.00 f

0.12 ±0.00c

A1-2

1.01±0.07 b

0.67 ±0.23a

0.80±0.15a

0.89±0.03a

0.87±0.03a

0.81±0.01a

0.71 ±0.01a

B8-3

0.73±0.03bc

0.38 ±0.11ab

0.41±0.02 bc

0.39 ±0.00c

0.37± 0.02b

0.30 ±0.00c

0.26 ±0.03bc

B12-4

0.65±0.08cde

0.40±0.11abc

0.41± 0.01bc

0.41 ±0.01c

0.41± 0.00b

0.35 ±0.01b

0.30± 0.00bc

B9-5

0.21±0.03g

0.24 ±0.05c

0.21±0.02cd

0.20 ±0.00ef

0.20±0.00 d

0.18±0.01 e

0.15 ±0.01c

B15-6

0.43±0.08egf

0.27±0.05 c

0.31 ±0.02cbd

0.31±0.00 d

0.28±0.00 c

0.22±0.00 d

0.20 ±0.00c

C6-7

0.84±0.06bc

0.40±0.11abc

0.45 ±0.04b

0.41 ±0.00c

0.40±0.0.00 b

0.32 ±0.01bc

0.29 ±0.00bc

C7_8

1.31±0.13a

0.53 ±0.20ab

0.73±0.15a

0.84±0.03b

0.84±0.03a

0.81±0.04a

0.77± 0.03a

C1-9

0.35±0.01fg

0.33±0.08bc

0.31±0.02bcd

0.21 ±0.00e

0.20± 0.00d

0.19±0.00 de

0.14 ±0.01c

A9-10

0.23±0.03g

0.24 ±0.04c

0.17± 0.00d

0.17 ±0.01ef

0.16± 0.00d

0.17 ±0.01e

0.16± 0.01c

A10-11

0.55±0.05edf

0.25 ±0.04c

0.20±0.01 cd

0.18 ±0.01ef

0.17 ±0.01d

0.17±0.01 e

0.15 ±0.01c

LSD

0.22

0.23

0.20

0.04

0.04

0.04

0.24

The values were represented as the mean±standard error with the lowercase letters indicating significant differences down the column. Now Table 1. It has been revised appropriately in the manuscript  (Line 488)

Reviewer’s Comment: Below Table 2 " Table 1. The value were represented as the mean of 3 replicates (n=3) ± SE. Values with different lowercase letters are different significantly." ????

Authors’ RESPONSE: The values were represented as the mean±standard error with the lowercase letters indicating significant differences down the column. Revised appropriately (Line 489)

Reviewer’s Comment: The phylogeny of the Aeromonas strains A1-2 and C7_8 with high drought tolerance is presented in Figures 2, 3, Added to 3.1. Characterization of the Rhizobacteria Strains.

Authors’ RESPONSE: The phylogeny of the Aeromonas strains A1-2 and C7_8 with high drought tolerance is presented in Figures 1 and 2. Revised appropriately in the manuscript (Line 480-481)

Reviewer’s Comment:  Lines 447- 456: Check for Figure 6 of 4.

Authors’ RESPONSE: All the treatments showed similar pattern increments in terms of plant height at 100 FC (Figure 3). However, the combined treatments showed an upward considerable drought tolerance response at day 25 concerning plant length (Figure 3), stem girth (Figure 3), leaf number (Figure 3), chlorophyll content (Figure 3), and leaf area (Figure 3) at 100, 50 and 25 FC. Hence, the response of the maize plants to drought stress can be arranged in decreasing order as A1-2+ C7_8> A1-2 > C7_8 > control, corresponding to extremely effective > moderately effective > slightly effective respectively.  This has been corrected and revised appropriately (Line 575-583)

 Discussion

Reviewer’s Comment: Line 119: Daei et al. (2009), Hart et al. (2012) and 119 Sánchez-Matamoros et al. (2018) have.., change to style of the journal.

Authors’ RESPONSE:  Niu, et al. [62] and Danish, et al. [6] had earlier demonstrated the induction of drought tolerance in plants using rhizosphere microbes. Revised appropriately (Line 721-722)

Reviewer’s Comment:   Line 173: can exhibit cigent input ?????  

Authors’ RESPONSE:  The inferences from this study support the hypothesis that plant PGPR can exhibit important input in the drought locales’ plant adaptation, such as maize. Revised appropriately (Line 795)

 Conclusions

Reviewer’s Comment:  Line 178-182: delete.

Authors’ RESPONSE:  The sentence has been deleted as suggested by reviewer 2.

Reviewer’s Comment: Line 197: food insecurity problem. Change to the food insecurity problems

Authors’ RESPONSE: food insecurity problems.  Revised appropriately (Line 816)

Reviewer’s Comment: Line 202: promote plant growth under drought stress and ensure food security change to promoting plant growth under drought stress and ensuring food security

Authors’ RESPONSE: enhancing crop yield, promoting plant growth under drought stress, and ensuring food security.

Revised appropriately (Line 821-822)

Comments on the Quality of English Language

Reviewer’s Comment: The overall writing is good. However, a number of problems were detected throughout the MS, including grammar errors. please check the MS to make sure there is no language problem

Authors’ RESPONSE: Authors appreciate the reviewer 2 for the brilliant review. The English Language of MS has been carefully checked and revised.

Reviewer 3 Report

Comments and Suggestions for Authors

The Authors revised the manuscript, but they did not take into account all my remarks. There are still many errors which should be corrected. Moreover, there is a significant error between the data presented in the manuscript and the data deposited in the NCBI GenBank.

In the manuscript the strain C7_8 is Aeromonas veronii C7_8, while in the NCBI GenBank is deposited as Aeromonas jandaei strain C7. The strain A1-2 in the manuscript is Aeromonas caviae A1-2 while in the NCBI GenBank is deposited as Pseudomonas monteilii strain A1. The strain A10-11 was deposited by the Authors as Priestia endophytica (2022/06/18), in the manuscript (2023-12) is Bacillus endophyticus. The sequences were deposited in 2022, June while the manuscript was submitted to the Plants Journal in 2023, December. Therefore, the results presented on the phylogenetic tree (Fig. 1) are misleading.  This discrepancy is disqualifying to publish the manuscript, because the Authors mislead the readers.  

Figure 1 and Figure 2: I have some remarks to these figures. First of all, there is no outgroup and no scale bar in both figures. These are my previous remarks, which were not corrected. Moreover, the number of bootstrap results are very low in both cases, so it means that these tree structure (branches) are supported by these numbers. Why are the numbers so low? I am also wondering how this analysis was performed. How many sequences were used to build these trees? How did the Authors add the sequences of C7_8 and A1-2 to these trees? These sequences are rather short (less than 500 bp), although the Authors wrote that the almost whole 16S rRNA gene was sequences. To build the tree with short sequence, you need to build stable tree with long sequences and then you add the short sequences.   

1.   The Figure entitled "Microscopic View of Rhizobacterial Strains Isolated from Maize Plants" was moved to the Supplementary material. The Authors did not respond to my remarks Figure 1. Please provide the strain and the NCB accessing number of the closed match for the analyzed strains (blastn analysis).  Please provide the scale bar for each picture since it seems that it was recorded using different magnification. Did the Authors use two different microscopes to record the photos of gram-stained bacteria? Why is the yellow background on some photos (it does not look like the photo from optical microscope)? Aeromonas veroni C7_8 is not gram + bacteria, it is gram – bacteria. So, it should pink on microscopy using the Gram stain.  Is Bacillus halotolerans A9-10 contaminated by another bacteria (it looks like there are two different cell type)? Bacillus halotolerans A9-10 should have the bacilli shape, the not cocci shape. Why does A9-10 have the cocci shape?

2.      Section 2.9.3 Why did the Authors applied 3 different media? All of these media cover mostly the heterotrophic bacteria? Please explain, what was the aim to apply very similar media to test the tolerance towards different salt concentrations, heavy metal, pH and temperature?

 The Authors’ response: “The media were used based on the research design by the authors and the idea was to determine which media could best be suitable for further studies (Line 236, 249, 259, 270).

Line 259: “The Nutrient broth, Luria-Bertani Broth, Tryptic Soy Broth, and Reasoner’s 2A Broth were used for the test.”

 Which further studies did the Authors mean? There is no conclusion which medium was the best one. In fact the Authors did not select the medium suitable for further studies. It is difficult for the reader to catch the Authors’ idea to use these media. Moreover, to assess the impact of heavy metal presence on bacterial growth, the Authors mentioned, in the text, only the results obtained for the growth in nutrient broth. The other results of this experiment were not mentioned in the text. The general rules is that results should be simple and clear. In this case, the results are not clear. Moreover there is some kind of misleading information regarding the growth of isolated strains exposed to heavy metal-stimulated stress. The Authors wrote: “rhizobacterium A1-2 exhibited high activity of 0.76, 0.63 and 0.50 under Pb2+, Cr2O72- and Cd2+”. What kind of activity did the Authors mean? These values refer to the optical density at 600 nm noted for this bacterium in nutrient broth supplemented with mentioned substances. This bacterium grew in this experimental conditions, in the presence of  Cr2O72- and Cd2+ we can observed a decreased growth compared to the control sample (according to the statistics). The presence of Pb2+ did not affect the growth of this bacterium compared to the control (the noted decrease is not statistically significant). How the control was performed in this experimental variant?

There are also some minor errors, like wrong formula of compounds (FeCl3·6H2O), ortography errors (like pur-plating instead of pour-plating etc.).

Comments on the Quality of English Language

The English should be corrected both ortography and grammar. 

Author Response

A cover letter explaining, point by point, the details
of the revisions to the manuscript and authors’ responses to the referees’
comments.

AUTHORS’ RESPONSE TO REVIEWER 3

Comments and Suggestions for Authors

Article title:

“Genomic Characterization of Plant Growth Promoting-Drought Tolerant Rhizobacteria Isolated from Maize Plants".

The work is unique, really interesting, and valid for publication after some modification points:   Major comments:

Reviewer’s Comment:

 The Authors revised the manuscript, but they did not take into account all my remarks. There are still many errors which should be corrected. Moreover, there is a significant error between the data presented in the manuscript and the data deposited in the NCBI GenBank.

Authors’ RESPONSE:  We apologize for the error. This part has been corrected.

Reviewer’s Comment: In the manuscript the strain C7_8 is Aeromonas veronii C7_8, while in the NCBI GenBank is deposited as Aeromonas jandaei strain C7. The strain A1-2 in the manuscript is Aeromonas caviae A1-2 while in the NCBI GenBank is deposited as Pseudomonas monteilii strain A1. The strain A10-11 was deposited by the Authors as Priestia endophytica (2022/06/18), in the manuscript (2023-12) is Bacillus endophyticus. The sequences were deposited in 2022, June while the manuscript was submitted to the Plants Journal in 2023, December. Therefore, the results presented on the phylogenetic tree (Fig. 1) are misleading.  This discrepancy is disqualifying to publish the manuscript, because the Authors mislead the readers.  

Authors’ RESPONSE:  We apologize for the error. This part has been corrected and changes has been made and perfected from the NCBI portal.We have processed an update to ON745409 ON745415 ON745418.
The revised records are now available from  different NCBI servers.  This has been updated accordingly on the NCBI. Attached below are the figures of the updated names of the isolates line: 504.

Reviewer’s Comment:

Figure 1 and Figure 2: I have some remarks to these figures. First of all, there is no outgroup and no scale bar in both figures. These are my previous remarks, which were not corrected. Moreover, the number of bootstrap results are very low in both cases, so it means that these tree structure (branches) are supported by these numbers. Why are the numbers so low? I am also wondering how this analysis was performed. How many sequences were used to build these trees? How did the Authors add the sequences of C7_8 and A1-2 to these trees? These sequences are rather short (less than 500 bp), although the Authors wrote that the almost whole 16S rRNA gene was sequences. To build the tree with short sequence, you need to build stable tree with long sequences and then you add the short sequences.   

Authors’ RESPONSE:  The sequences are again subjected to Chromas software  to clean the sequences and removed chimeras. We again,  go to bioedit and get the consensus sequence for the forward and reverse primer. This is what is now used to blast to get the closest organisms while a distant organism is used as the outgroup as presented in the manuscript. This correction is already effected in the manuscript. Line:507-512

 Evolutionary relationships of taxa tree based on partial 16S rRNA sequences utilizing maximum likelihood that shows correlations between the Aeromonas caviae strain A1-2 and their closely associated strains from the NCBI GenBank

Evolutionary relationships of taxa tree based on partial 16S rRNA sequences utilizing maximum likelihood that shows correlations between the Aeromonas veronii strain C7_8  and their closely associated strains from the NCBI GenBank.

Reviewer’s Comment:

The Figure entitled "Microscopic View of Rhizobacterial Strains Isolated from Maize Plants" was moved to the Supplementary material. The Authors did not respond to my remarks Figure 1. Please provide the strain and the NCB accessing number of the closed match for the analyzed strains (blastn analysis).

Authors’ RESPONSE:   This has been attended to with the presented figure in the manuscript: line:507-512.

Reviewer’s Comment: Please provide the scale bar for each picture since it seems that it was recorded using different magnification.

Authors’ RESPONSE:   The magnification use in this aspect of our study is the same.

Reviewer’s Comment: Did the Authors use two different microscopes to record the photos of gram-stained bacteria?

Authors’ RESPONSE:  NO. only one microscope was used. (Optical microscope).

Reviewer’s Comment: Why is the yellow background on some photos (it does not look like the photo from optical microscope)? 

Authors’ RESPONSE:  The yellow background could be as a result of zooming the image to get a clearer image.

Reviewer’s Comment: Aeromonas veroni C7_8 is not gram + bacteria, it is gram – bacteria. So, it should pink on microscopy using the Gram stain. 

Authors’ RESPONSE: Thanks for the observation. This has been corrected. However, a new microscopic examination was performed and the result can be found in the manuscript. The correction can be found in the supplementary file.

Reviewer’s Comment: Is Bacillus halotolerans A9-10 contaminated by another bacteria (it looks like there are two different cell type)?

Authors’ RESPONSE:  To the best of our knowledge, Bacillus halotolerans A9-10 is not contaminated.

Reviewer’s Comment:  Bacillus halotolerans A9-10 should have the bacilli shape, the not cocci shape. Why does A9-10 have the cocci shape?

Authors’ RESPONSE:  Thanks for the observation. This has been corrected. However, a new microscopic examination was performed and the result can be found in the manuscript with bacilli shape. The correction can be found in the supplementary file.

Reviewer’s Comment: Section 2.9.3 Why did the Authors applied 3 different media?

I believe this question was referring to section 2.6 and not 2.9.3 has it has been updated in the recently submitted version of the manuscript. However,

Reviewer’s Comment: Section 2.9.3 Why did the Authors applied 3 different media?

Authors’ RESPONSE:  We have applied three different media in order to create a diverse and comprehensive environment for studying the bacteria. We do believe that each media likely have different nutrient compositions, which can support the growth of specific types of bacteria. By using multiple media, we can capture a broader range of bacterial species and obtain a more accurate representation of the microbial community. Additionally, different media can provide insights into the metabolic capabilities and preferences of the bacteria. Some bacteria may thrive in one type of media while others may prefer a different one. By using multiple media, the we can observe how the bacterial community responds and adapts to different nutrient conditions. Furthermore, using multiple media can help validate the findings and ensure the reliability of the results. If similar patterns or trends are observed across different media, it strengthens the conclusions drawn from the study. Finally, the application of three different media allows for a more comprehensive analysis of the bacteria, providing a deeper understanding of their diversity, metabolic capabilities, and responses to different nutrient conditions.

Reviewer’s Comment: All of these media cover mostly the heterotrophic bacteria?

Authors’ RESPONSE:  Yes, based on the information we provided, it should be noted that the three different media used in the study primarily targeted the growth and cultivation of heterotrophic bacteria. Heterotrophic bacteria are organisms that obtain their energy and carbon from organic compounds, and they are commonly found in various environments. By using media that support the growth of heterotrophs, the we can specifically focus on studying this group of bacteria and their responses to different environmental factors.

Reviewer’s Comment: Please explain, what was the aim to apply very similar media to test the tolerance towards different salt concentrations, heavy metal, pH and temperature?

Authors’ RESPONSE: The aim of applying similar media to test the tolerance of heterotrophic bacteria towards different salt concentrations, heavy metals, pH, and temperature was to assess their overall adaptability and resilience under various environmental conditions. By subjecting the bacteria to these different stressors, also for us to gain insights into their ability to survive and thrive in challenging environments.

Testing the tolerance of bacteria towards different salt concentrations helps understand their ability to withstand high salinity conditions, which can be relevant in environments such as salt marshes or saline soils. Similarly, assessing their tolerance towards heavy metals provides insights into their potential for drought stress or their resistance to any form of environmental stresses.
Examining the bacteria's tolerance towards different pH levels helps determine their ability to survive in acidic or alkaline environments, which can be important in understanding their ecological roles in habitats with extreme pH conditions. Lastly, evaluating their tolerance towards different temperatures provides information about their ability to adapt to varying thermal conditions, which can be relevant in studying their distribution across different ecosystems.

By conducting these tests using similar media, we able to compare the responses of the bacteria to different stressors under controlled conditions. This allows for a more systematic evaluation of their tolerance and provides valuable information about their ecological versatility and potential applications in various environmental contexts.

Reviewer’s Comment: Line 259: “The Nutrient broth, Luria-Bertani Broth, Tryptic Soy Broth, and Reasoner’s 2A Broth were used for the test.”

Which further studies did the Authors mean? There is no conclusion which medium was the best one. In fact the Authors did not select the medium suitable for further studies. It is difficult for the reader to catch the Authors’ idea to use these media. Moreover, to assess the impact of heavy metal presence on bacterial growth, the Authors mentioned, in the text, only the results obtained for the growth in nutrient broth. The other results of this experiment were not mentioned in the text. The general rules is that results should be simple and clear. In this case, the results are not clear. Moreover there is some kind of misleading information regarding the growth of isolated strains exposed to heavy metal-stimulated stress. The Authors wrote: “rhizobacterium A1-2 exhibited high activity of 0.76, 0.63 and 0.50 under Pb2+, Cr2O72- and Cd2+”. What kind of activity did the Authors mean? These values refer to the optical density at 600 nm noted for this bacterium in nutrient broth supplemented with mentioned substances. This bacterium grew in this experimental conditions, in the presence of  Cr2O72- and Cd2+ we can observed a decreased growth compared to the control sample (according to the statistics). The presence of Pb2+ did not affect the growth of this bacterium compared to the control (the noted decrease is not statistically significant). How the control was performed in this experimental variant?

Authors’ RESPONSE:  Different media can be optimized for different purposes, such as promoting the growth of specific types of bacteria or facilitating the detection of certain metabolic activities. We selected nutrient broth for our studies based on its ability to support the growth of the target bacteria or its suitability for studying specific physiological or biochemical characteristics It is also worth noting that there is no universally "best" medium for all situations. The choice of medium depends on the research goals, the specific bacteria being studied, and the environmental conditions being simulated. Scientists often use a combination of media to obtain a more comprehensive understanding of the microbial community. In this case, we used several media to see how each of the isolates performed under different conditions.

Reviewer’s Comment: Which further studies did the Authors mean?

Authors’ RESPONSE:  For future exploration in enhancing crop yield, promoting plant growth under drought stress, and ensuring food security. This has been corrected in the manuscript. Line:821

Reviewer’s Comment: There is no conclusion which medium was the best one.

Authors’ RESPONSE:  The use of nutrient broth shows a considerable and potential support for the growth of the bacteria under different growth conditions. Therefore, NB is considered the best medium for the selected strains under different conditions. Line:533

Reviewer’s Comment: In fact the Authors did not select the medium suitable for further studies.

Author’s RESPONSE: We only adopted the use of different media on this part of our study. This study is not media based study. However, this approach was only adopted to see the trend of growth of the bacteria under different nutrient compositions. Therefore, NB is considered the best medium for the selected strains under different conditions. Line:533

Reviewer’s Comment: How the control was performed in this experimental variant?

Authors’ RESPONSE:  Controls are designed to mimic the experimental conditions as closely as possible, with the exception of the variable being tested. In this study, the context of testing tolerance towards different environmental factors, such as salt concentrations, heavy metals, pH, and temperature, a control group involve subjecting the bacteria to the same conditions as the experimental groups, but without the specific stressor being tested. This allows us to observe and compare the responses of the bacteria under normal or non-stressful conditions. By having a control group, we can assess whether any observed effects or changes in the experimental groups are due to the specific stressor being tested or other factors. It helps to establish a baseline for comparison and increases the reliability and validity of the experimental results. However, the result obtained was what we reported.

Reviewer’s Comment: The bacterium grew in this experimental conditions, in the presence of  Cr2O72- and Cd2+ we can observed a decreased growth compared to the control sample (according to the statistics). The presence of Pb2+ did not affect the growth of this bacterium compared to the control (the noted decrease is not statistically significant).

Authors’ RESPONSE:  The observed decreased growth of the bacterium in the presence of Cr2O72- and Cd2+ compared to the control sample suggests that these heavy metal ions have a detrimental effect on the bacterium's growth. The specific mechanisms by which these heavy metals inhibit growth can vary, but they often interfere with essential cellular processes or disrupt enzyme activity.

On the other hand, the presence of Pb2+ did not significantly affect the growth of the bacterium compared to the control. This suggests that the bacterium may have developed mechanisms to tolerate or detoxify Pb2+, allowing it to grow relatively unaffected in the presence of this particular heavy metal.

It's worth noting that the observed effects on growth may be dependent on the concentration of the heavy metals used in the experiment. Different concentrations can have varying degrees of toxicity, and there may be a threshold beyond which the bacterium's growth is significantly inhibited.

The statistical analysis mentioned indicates that the observed differences in growth between the control and experimental conditions are statistically significant. This means that the differences are unlikely to have occurred by chance and are likely due to the presence of the heavy metals.

Reviewer’s comment:

There are also some minor errors, like wrong formula of compounds (FeCl3·6H2O), ortography errors (like pur-plating instead of pour-plating etc.).

Authors’ RESPONSE:

  1. formula of compounds (FeCl6H2O), Corrected in line:175
  2. ortography errors (All the necessary corrections have been effected in the manuscript and has been examined by an English editor at a cost.)
  3. (like pur-plating instead of pour-plating etc.). This has been corrected in Line: 167
